# Using Deep Learning to integrate paleoclimate and global biogeochemistry over the Phanerozoic Eon

Dongyu Zheng[1,2], Andrew S. Merdith[2,3], Yves Goddéris[4], Yannick Donnadieu[5], Khushboo Gurung[2], Benjamin J.W. Mills[2]

[1]State Key Laboratory of Oil and Gas Reservoir Geology and Exploitation and Key Laboratory of Deep-time Geography and Environment Reconstruction and Applications, MNR, Institute of Sedimentary Geology, Chengdu University of Technology, Chengdu, China
[2]School of Earth and Environment. University of Leeds. Leeds, UK
[3]School of Physics, Chemistry and Earth Sciences, University of Adelaide, Adelaide, Australia
[4]Géosciences Environnement Toulouse, CNRS–Université de Toulouse III, Toulouse, France
[5]Aix Marseille Univ, CNRS, IRD, INRAE, CEREGE, Aix-en-Provence, France

*Correspondence to*: Dongyu Zheng (zhengdongyu@cdut.edu.cn)

**Abstract.** Databases of 3D paleoclimate model simulations are increasingly used within global biogeochemical models for the Phanerozoic Eon. This improves the accuracy of the surface processes within the biogeochemical models, but the approach is limited by the availability of large numbers of paleoclimate simulations at different $pCO_2$ levels and for different continental configurations. In this paper we apply the Frame Interpolation for Large Motion (FILM) Deep Learning method to a set of Phanerozoic paleoclimate model simulations to upscale their time resolution from one model run every ~25 million years to one model run every 1 million year (Myr).

Testing the method on a 5 Myr time-resolution set of continental configurations and paleoclimates confirms the accuracy of our approach when reconstructing intermediate frames from configurations separated by up to 40 Myrs. We then apply the method to upscale the paleoclimate data structure in the SCION climate-biogeochemical model. The interpolated surface temperature and runoff are reasonable and present a logical progression between the original keyframes.

When updated to use the high-time-resolution climate datastructure, the SCION model predicts climate shifts that were not present in the original model outputs due to its previous use of wide-spaced datasets and simple linear interpolation. We conclude that a time resolution of ~10 Myr in Phanerozoic paleoclimate simulations is likely sufficient for investigating the long-term carbon cycle, and that Deep Learning methods may be critical in attaining this time-resolution at a reasonable computational expense, as well as for developing new fully-continuous methods in which 3D continental processes are able to translate over a moving continental surface in deep time. However, the efficacy of Deep Learning methods in interpolating runoff data, compared to that of paleogeography and temperature, is diminished by the heterogeneous distribution of runoff. Consequently, interpolated climates must be confirmed by running a paleoclimate model if scientific conclusions are to be based directly on them.

## 1 Introduction

To simulate global environmental change over Phanerozoic time it is important to understand how continental surface processes operate. For example, the weathering of silicate minerals controls the removal of atmospheric $CO_2$, and phosphorus input from weathering plays a major role in the long-term oxygenation of the Earth (Walker et al., 1981; Lenton and Watson, 2004). Weathering rates are largely controlled by local erosion rates, temperature and hydrology (West 2012; Maher and Chamberlain, 2014), and the latest generation of Phanerozoic global biogeochemical models aim to represent these factors at the local scale using data from 3D General Circulation Model (GCM) simulations (Godderis et al., 2023). Due to the long computational timescales of GCMs (typically weeks to months per ~5000-year simulation for a fully coupled ocean-atmosphere model), they cannot be run interactively with long-term biogeochemical cycles over millions of years. Therefore, the 'spatialized' deep-time biogeochemical models such as GEOCLIM (Donnadieu et al., 2006; Godderis et al., 2014) and SCION (Mills et al., 2021; Longman et al., 2022) rely on either discrete time intervals, or linear interpolation between times set by previously-computed climate model simulations.

Currently both the GEOCLIM and SCION models use a set of 22 continental configurations (including the present day) whose climate has been simulated by the Fast Ocean and Atmosphere Model (FOAM) at a range of different $CO_2$ levels. This equates to one set of model runs every ~25 million years on average, although some gaps are up to 55 million years. This coarse time resolution has likely impacted the accuracy of the biogeochemical model results. For example, through plate tectonic motion, a mountain range may pass through the tropics, an event expected to cause a spike in continental weathering due to high rainfall, but this may be undetected by SCION or GEOCLIM if the timespan at which the mountain range crossed the equator was not represented in the time points chosen for the paleoclimate simulations. A further issue is that when these models are focused on single events, such as mass extinctions, they may not be able to incorporate the relevant continental configurations and climate fields for that time in Earth history, instead using boundary conditions for up to 20 million years before or after the event.

Deep learning has received significant attention in the field of geosciences due to its impressive capabilities in handling tasks such as regression, classification, time-series analysis, and image processing (Reichstein et al., 2019; Chen et al., 2022; Zheng et al., 2022). One notable application of deep learning is in video frame interpolation, where it synthesizes intermediate frames between two input frames (Niklaus et al., 2017; Shi et al., 2022). Such a process can be highly beneficial in creating higher-time-resolution input variables for biogeochemical models by interpolating from the original climate model runs.

In this paper, we first performed a numerical and visual validation of the Deep Learning interpolation of a PaleoDEM topographic elevation dataset (Scotese and Wright, 2018), as well as surface air temperature generated from these maps using the HadCM3L GCM (Scotese et al., 2021; Valdes et al., 2021). The validation results suggest that the Deep Learning method is capable of adequately detecting plate motions and changes to surface air temperature. We then use Deep Learning to fill the gaps between the paleoclimate simulation set used in the SCION model, increasing the time-resolution around 25-fold to 1 million years. We focus on the SCION model because it runs continuously over the Phanerozoic and has previously published

outputs for long-term atmospheric $CO_2$, $O_2$ and global average temperature, but our results could also be used to produce new runs of the GEOCLIM model, as well as other Phanerozoic models that require spatial surface process information.

## 2 Data and methodology

### 2.1 Model forcings at 22 distinct time intervals

The SCION model employs a series of 2D model forcing fields taken from annual means of the FOAM climate model, which
were initially developed for the GEOCLIM model (Godderis et al., 2014). These fields are paleogeography (a composite of works by Blakey, Besse and Fluteau, and Sewall – see Godderis et al., 2014 for details), surface air temperature, continental runoff and topographic slope (Fig. 1). These 2D fields are 40×48 cells (4.5° latitude ×7.5° longitude) and are available for 22 distinct time points (time intervals shown in Fig. 6) roughly evenly-spaced between the Cambrian and present day. These 22 time points represent the grid-data stack times in the context of climate modelling. They are also run for a large range of
different $CO_2$ levels, and extrapolated beyond these for a total of 26 different levels by applying a linear interpolation; the FOAM surface air temperature and continental runoff are adjusted according to a linear change in the logarithm of $CO_2$ concentrations. During the SCION model run, 2D linear interpolation is used to estimate these fields for the current model $CO_2$ level, and a weighted mean is used to produce a final estimate of bulk weathering fluxes by using the distance between the current model timestep and the available climate model runs. Wide spacing in time between some of these climate model
datasets means that this weighted mean technique will likely miss many important features of Phanerozoic climate change. To improve on this, we adopt a frame interpolation technique (Reda et al., 2022), widely utilized to synthesize intermediate frames between two input frames in video sources, which increases the time-resolution (e.g. increases frames per second). This technique typically finds applications in amplifying refresh rates or generating slow-motion videos (Wu et al., 2023).

### 2.2 The deep learning interpolation algorithm

Deep Learning models are complex neural networks with typically $>10^6$ parameters. The model emulates the learning process of humans by updating the parameters in the neural networks to produce optimal results. The principal idea of using deep learning in frame (e.g., image) interpolation is to estimate the optical flow, which symbolizes the changes between two consecutive frames, followed by a pixel synthesis process that restores the intermediate images based on the estimated optical flows. The deep learning model aims to minimize the differences between model-predicted intermediate frames and the actual
intermediate frames in a training dataset. This approach enables the model to accurately extrapolate the visual transformation from one frame to the next and synthesize a plausible intermediate frame that maintains temporal consistency with the surrounding frames. The inputs of the deep learning model are two end-point image frames, with the target being the intermediate frames. By understanding how intermediate frames evolve from previous and future frames across a broad spectrum of video datasets, the deep learning model can discern rotation, scaling, colour changes, and more intricate
deformations, making it fit for creating interpolation images for new tasks.

In our study, we employ a complex Convolution Neural Network (CNN), called Frame Interpolation for Large Motion (FILM, Reda et al., 2022), to generate image interpolations for the FOAM dataset. FILM, when contrasted with traditional interpolation algorithms and other deep learning techniques, exhibits superior proficiency in dealing with abrupt changes in brightness and substantial motion. Such sudden shifts are frequent in the FOAM dataset due to the significant reorganizations that the paleogeographic and paleoclimatic conditions underwent throughout the Phanerozoic Eon (Royer et al., 2004; Godderis et al., 2014; Scotese, 2021). We represent the FOAM dataset directly from the native R15 climate grid as a set of 40×48-sized images, each depicting the entire Earth surface. The FILM technique is then deployed to generate intermediate images between two consecutive model forcing frames from a specific dataset. This method enables the creation of an atlas of one-million-year model forcing frames through several iterations of interpolation (See Fig. 1). By converting these interpolated frames back to numerical values, this atlas can be used to generate million-year-resolution 'DeepFOAM' dataset, which can then be used in the place of the original SCION model forcing set to run the SCION model. Nothing is altered at the SCION model runtime—it still uses weighted averages to interpolate bulk continental fluxes in time—however now it will interpolate between a maximum gap of 1 million years, where variations in the continental configurations are very small.

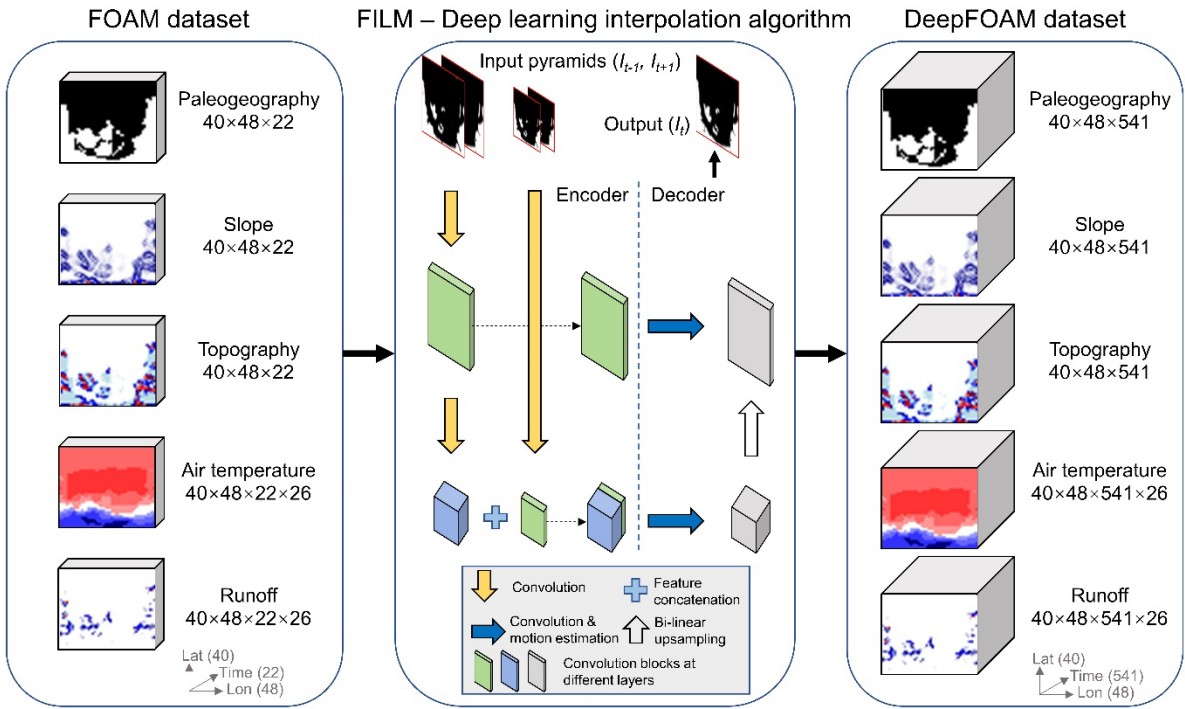

**Figure 1. Deep learning interpolation workflow for FOAM dataset and simplified overview of FILM.** Numerical annotations represent the dataset dimensions, e.g., for original air temperature, 40 and 48 refer to the horizontal and vertical geographic spans, 22 represents the time intervals, 26 represents the different $CO_2$ concentration levels. $I_{(t-1)}$ and $I_{(t+1)}$ represent the two input frames; $I_t$ is the intermediate frame. The convolution step is the matrix operation to obtain high-level feature representations; motion estimation is the change estimation achieved by the convolution operation; concatenation means the unification of two matrices of the same dimension to amalgamate features from different levels in the input pyramids; bi-linear up sampling is the technique used to reconstruct a frame from the high-level feature

representation. The FILM overview is simplified based on Fig. 2 in Reda et al. (2022); we refer the reader to Reda et al. (2022) and Goodfellow et al. (2016) for further details of these techniques.

FILM is a U-shaped CNN encompassing both encoders and decoders (Fig. 1). Convolution, the fundamental operation in any CNN, involves multiplication between the image matrix and the filters, which are typically smaller matrices. This convolutional operation yields a summarized representation of the original images. Deep CNNs employ hundreds or thousands of such filters to discern features in images such as shape, brightness and patterns (Hinton et al., 2006; Goodfellow et al., 2016). The encoder module in FILM serves to extract high-level feature representations from input images. This is

accomplished via its specialized pyramidal architecture that encompasses seven levels of feature extractors. These range from fine-level (high-resolution) to coarse-level (low-resolution), effectively allowing FILM to detect both fine and coarse changes in the images, with each successive extractor operating on an input frame with half the resolution of the previous level's inputs. These input frames then undergo several layers of convolution blocks to extract high-level feature representations and bi-directional motion between the input and interpolated frames. The feature representations, coupled with the bi-directional

motion, synthesize the high-level representations of the intermediate frame. The decoder module of FILM then uses these high-level representations to reconstruct the intermediate images. During the training process, the weights in the convolution blocks within both the encoder and decoder are continually adjusted to minimize the differences between the model-predicted images and the training images. Having undergone training with over 100,000 unique videos, the pre-trained FILM model exhibits the capability to discern rotation, scaling, colour changes, movements, and more intricate deformations, making it

ideally suited for creating interpolation images for novel tasks.

We use the pre-trained FILM model to create interpolated images without conducting additional training. By running the FILM model with our image dataset (comprising 22 images for a given $CO_2$ concentration at each time-interval, n, it yields 21, n-1, interpolation images. During the second round of prediction, both the original and interpolated images are used to generate further interpolation images between the original images and the first-round interpolation images. By iterating this operation

k times, $(2^{k-1})\times(n-1)$ interpolation images are generated in total, and $2^{k-1}$ images are formed between each of the two original images. Given that the maximum age gap between the original model forcing is 55 Ma, we executed 6 iterations of prediction to produce 63 interpolation images between each pair of original images and selected at most 54 out of the 63 images to represent the dataset for each million year time point. These selections were made evenly across the 63 images to ensure a uniform and representative sampling for each million year in the dataset.

**3 Validating the method and interpolated datasets**

**3.1 Validation of interpolation using a PaleoDEM dataset**

While the FILM model has demonstrated a robust ability to interpolate complex changes between input and intermediate frames, largely due to extensive training on over 100,000 videos, its performance has not yet been scrutinized in the context

of paleogeographic and paleoclimate datasets. To test this application, we first apply the FILM model to a high-time-resolution paleo-digital elevation dataset (PaleoDEM, Scotese and Wright, 2018) that delineates the evolving distributions of land and oceans over the past 540 million years in five million-year intervals. By partitioning the PaleoDEM dataset into input and intermediate frames, we could contrast the FILM-predicted frames with the actual intermediate frames, quantifying their disparities. The similarity between the predicted and actual frames serves as a testament to FILM's efficacy in capturing plate movements and transformations.

The PaleoDEM dataset comprises 109 files, each file includes estimations of land surface elevation and ocean depth, measured in meters, at a resolution of 1×1 degrees. Hence, the PaleoDEM is a 361× 181 dimensional dataset where 361 represents longitude (ranging from -180 to 180) and 181 denotes latitude (ranging from -90 to 90). To make the PaleoDEM comparable with our 48×40 dimensional dataset, we applied nearest-neighbour interpolation, a downsampling algorithm, to downscale the PaleoDEM resolution to 48×40 by assigning the values of the closest pixel to the new pixel locations. Moreover, any location with elevation values greater than zero was characterized as land (denoted by 255 in pixel values), with the remainder classified as oceans (denoted by 0 in pixel values).

The PaleoDEM was partitioned into distinct input and output datasets using temporal intervals of 10 million years (Myrs), 20 Myrs, and 40 Myrs. This strategy facilitated three separate validation procedures. In the first validation approach, we use a 10 Myr interval. The PaleoDEM sequences from 540 million years ago (Ma), 530 Ma, etc., up to 0 Ma, were designated as the input dataset. Correspondingly, the sequences from 535 Ma, 525 Ma, and so forth, until 5 Ma, were selected as the output dataset which the FILM model outputs will be compared against. For the second validation process, we adopted a 20 Myr interval. This time, the input dataset comprised PaleoDEM sequences from 540 Ma, 520 Ma, etc., down to 0 Ma. The sequences from 535 Ma, 530 Ma, 525 Ma, etc., to 5 Ma served as the output dataset. An identical procedure was executed for the third validation scheme, but with a 40 Myr interval. In each case we use the input dataset to make interpolation frames using FILM, and by comparing the predicted frames with the real frames in the output dataset, the model's predictive accuracy can be assessed.

For a systematic and comprehensive investigation of the FILM model performance, we calculate the Structural Similarity Index (SSIM), Peak Signal-to-Noise Ratio (PSNR), two-dimensional correlation and Normalized Root Square of the Mean Square Error (NRMSE), which are the most widely utilized performance measurements for frame interpolation (Wang et al., 2004; Dong et al., 2023). These widely accepted metrics can help us gauge differences between the actual intermediate frames and the predicted frames. SSIM is a metric to detect perceived changes that takes into account luminance, contrast, and structural information of the image. Given the real intermediate frame $I^R(x, y)$ and the FILM-predicted frame $\hat{I}(x, y)$, the SSIM is defined as:

$$\text{SSIM} = \frac{\left(2\mu_{\hat{I}}\mu_{I^R} + c_1\right) \times \left(2\sigma_{\hat{I}I^R} + c_2\right)}{\left(\mu_{\hat{I}}^2 + \mu_{I^R}^2 + c_1\right) \times \left(\sigma_{\hat{I}}^2 + \sigma_{I^R}^2 + c_2\right)}, \tag{1}$$

where $\mu_{\hat{I}}$ and $\mu_{I^R}$ are the mean of $\hat{I}$ and $I^R$, $\sigma_{\hat{I}}^2$ and $\sigma_{I^R}^2$ are the variance of the $\hat{I}$ and $I^R$, $\sigma_{\hat{I}I^R}$ is the covariance between $\hat{I}$ and $I^R$, $c_1$ and $c_2$ are constants to avoid instability when the denominator is close to zero. Values of SSIM ranges from -1 to 1, representing inversely identical and identical images, respectively.

The PSNR is a ratio between the maximum possible power of the image and the power of corrupting noise that affects the fidelity of the image's representation, it is defined as:

$$\text{PSNR} = 10 \times log_{10} \times \left( \frac{L^2}{\frac{1}{N} \sum_{x,y}^{N} \left( I^R(x,y) - \hat{I}(x,y) \right)^2} \right), \tag{2}$$

where $L$ is the maximum pixel values (255 for our images), $N$ is the number of pixels in the image. The greater the value of PSNR, the better the performance of the frame interpolation.

The two-dimension correlation is defined as the Pearson correlation coefficient calculated over the two dimensions of $I^R(x,y)$ and $\hat{I}(x,y)$. Values of two-dimension correlation are between -1 and 1, where 1 indicates the two images are identical, 0 means

the images are uncorrelated, and -1 means the images are inversely identical. The NRMSE measures the differences in pixel values between the $I^R(x,y)$ and the $\hat{I}(x,y)$. In contrast to the other metrics, lower NRMSE values indicates better performance (range between 0 and 1).

$$\text{NRMSE} = \frac{\sqrt{\frac{1}{N} \sum_{x,y}^{N} \left( I^R(x,y) - \hat{I}(x,y) \right)^2}}{I^R(x,y)_{max} - I^R(x,y)_{min}}, \tag{3}$$

Figure 2 and Table 1 detail the quantitative assessments derived from our implemented numerical metrics, indicating the

195 performance of the FILM technique when applied to the PaleoDEM dataset. During the validation phase, using a 10 Myr interval, the frames predicted by FILM demonstrated remarkable congruity with the actual frames. This is evidenced by the high values of SSIM, PSNR, and two-dimensional correlation, and a low value of NRMSE. The SSIM and 2D-correlation maintain consistently high values throughout the entire time span from 540 Ma to 0 Ma. This highlights the consistent performance of FILM, and its capacity to capture the paleogeographic reorganization in the PaleoDEM dataset across

Phanerozoic timescales.

Despite good performance in general, the PSNR and NRMSE depict a trend over time, with PSNR values decreasing and NRMSE values increasing over time – both suggesting a poorer fit to the real intermediate frames at time points closer to the present day. The observed trend is somewhat expected as the maps closer to the present day – which can draw on larger geological evidence bases – tend to exhibit more fine scale features, such as land patches depicted as only one or a few pixels

in the frames (Fig. 3). During frame interpolation, fine scale features such as minor land patches are more easily overlooked. Consequently, metrics like PSNR and NRMSE, which quantify the pixel discrepancies between the predicted and actual frames, underscore this pattern of detail loss.

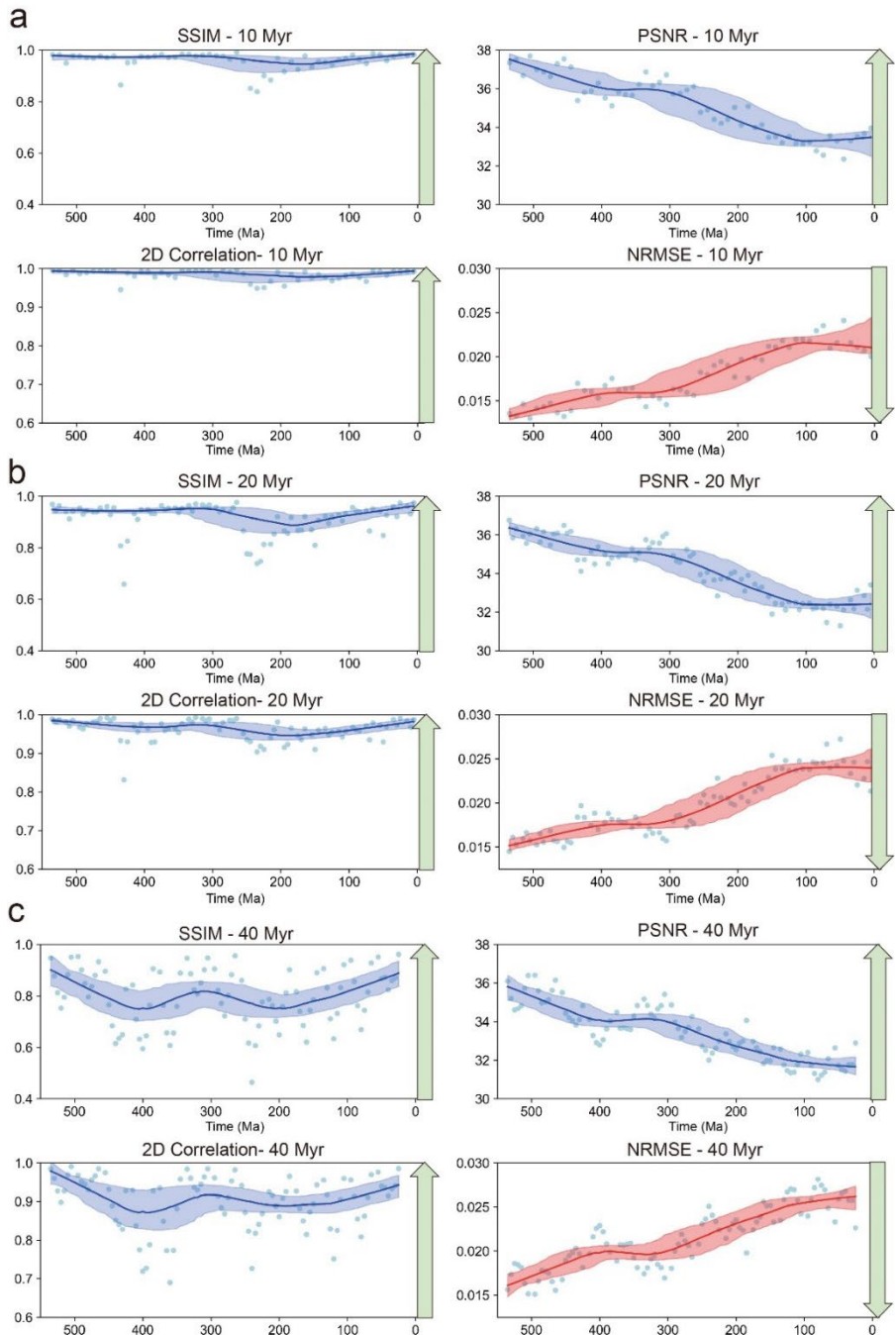

**Figure 2. Comparative Evaluation of Performance Utilizing (a) 10 Ma, (b) 20 Ma, and (c) 40 Ma Intervals within the PaleoDEM Dataset.** The blue line represents the LOWESS (Locally Weighted Scatterplot Smoothing) fitting curve with a fraction of 0.4, serving as an indicator of the central trend. The light blue shaded bands illustrate the confidence interval, derived from a 1000-resampling bootstrap method, providing a measure of the precision and uncertainty of the estimated fit. The NRMSE is represented in red colour because it signifies error, thus its trend is converse to the other three metrics. The green arrows indicate the direction of better performance. See text for detailed discussions.

**Table 1. Numerical evaluation on the PaleoDEM dataset**

| | 10 Myr | | | 20 Myr | | | 40 Myr | | |
|---|---|---|---|---|---|---|---|---|---|
| | Mean | Median | SD | Mean | Median | SD | Mean | Median | SD |
| SSIM | 0.96 | 0.97 | 0.03 | 0.91 | 0.94 | 0.06 | 0.80 | 0.81 | 0.11 |
| PSNR | 35.12 | 35.25 | 1.50 | 34.17 | 34.50 | 1.47 | 33.46 | 33.49 | 1.38 |
| 2D correlation | 0.98 | 0.99 | 0.01 | 0.96 | 0.97 | 0.03 | 0.90 | 0.93 | 0.07 |
| NRMSE | 0.02 | 0.02 | 0.00 | 0.02 | 0.02 | 0.00 | 0.02 | 0.02 | 0.00 |

SD, standard deviation. See section 3.1 for other abbreviations.

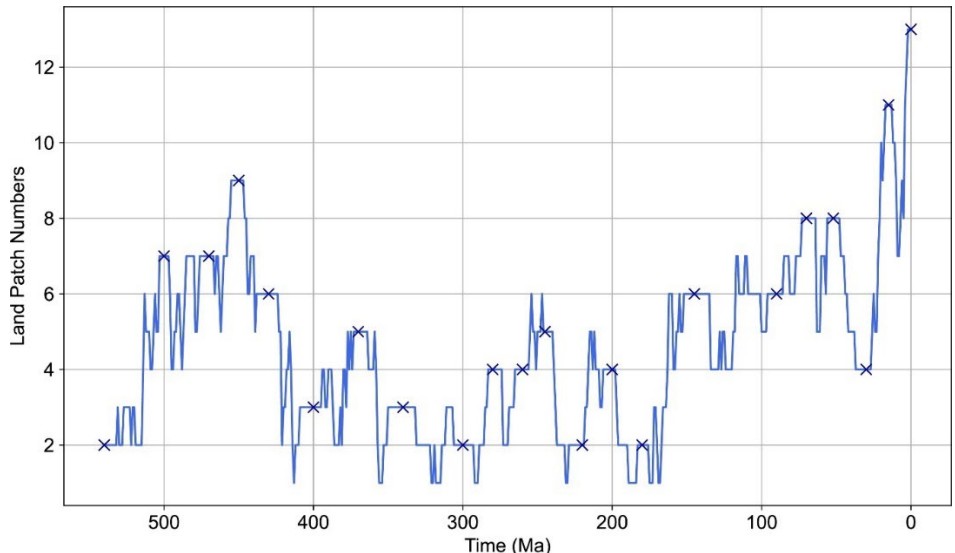

**Figure 3. Temporal evolution of land patch density using PaleoDEM from 540 Ma to present.** This figure indicates an overall increase of land patch density in the PaleoDEM dataset, attributed to the inclusions of more details in the more recent frames. This increasing trend may account for the patterns observed in PSNR and NRMSE. A land patch is defined as a contiguous grouping of land pixels. The density is calculated by dividing the number of land patches by the total frame size (40×48). The dark blue cross indicating the twenty-two time intervals used in the FOAM dataset.

For the comparison at 10 Myr and 20 Myr intervals, a high degree of similarity between the predicted and actual frames was discernible, as evidenced by the mean values of SSIM exceeding 0.8, 2D-correlation over 0.9, PSNR above 32, and NRMSE less than 0.025. For the 40 Myr interval validation, the mean values of SSIM were greater than 0.7, the mean values of 2D-correlation was above 0.80, the mean value of PSNR was over 31, and the mean values of NRMSE remained below 0.03. As anticipated, the performance deteriorates when the time interval is increased (Argaw and Kweon, 2022), which can be

attributed to the more significant changes between the two input frames. Interestingly, the SSIM and 2D-correlaiton show a particular decrease in performance around 220 Ma and 420 Ma. This may be due to more complex plate movements around these times which the algorithm finds more difficult to predict.

In addition to the numerical evaluation, we also performed visual inspections to detect obvious discrepancies between the original frames and the predicted frames. Across different time periods, predicted frames were generally visually comparable to original ones (see Fig. 2 for numerical estimations). The major deviations between the predicted and original frames were attributable to missing pixels or mismatched pixel values, and there were no changes in the placements of major land masses (Fig. 4). Given that the mean temporal interval for SCION model forcings is approximately 25 Ma, our numerical assessments and visual evaluations in comparing the FILM-predicted frames with the actual frames from the PaleoDEM dataset suggest that FILM possesses the capability to discern plate movements and transformations on a time scale appropriate to building interpolation frames for the SCION model forcings. Nevertheless, the FILM method creates a significant number of unmatched pixels compared to the original frames, which would alter climatic outputs of GCMs and linked biogeochemical calculations, especially as small introduced islands would be expected to have high runoff and chemical weathering rates (Park et al., 2020).

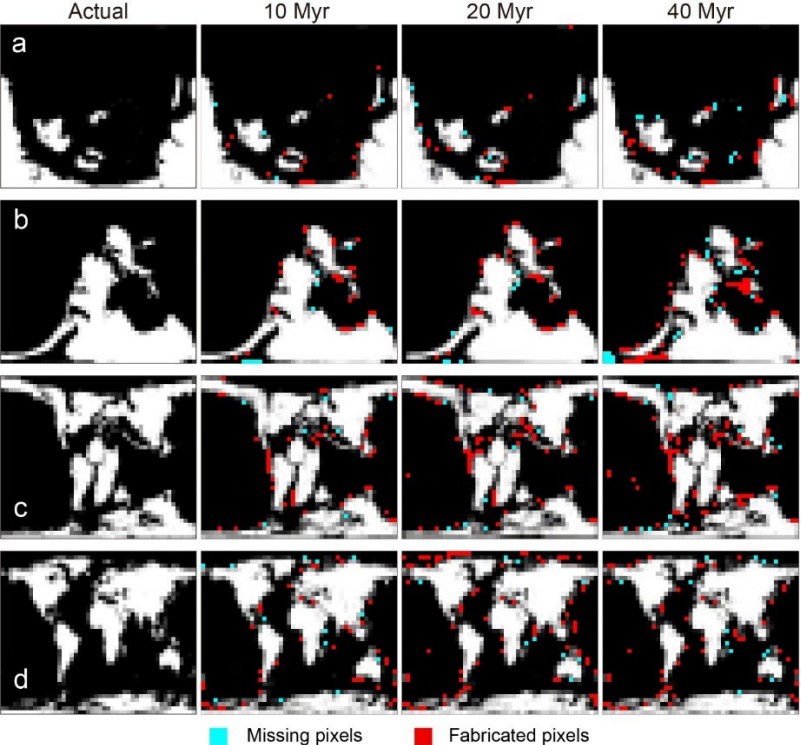

**Figure 4. Comparative visualization of actual PaleoDEM and model-predicted frames at selected time intervals.** This figure illustrates the comparative visualizations at (a) 535 Ma, (b) 315 Ma, (c) 105 Ma, and (d) 25 Ma. The grey pixels represent areas impacted by nearest-neighbour downsampling in the paleogeographic maps. The missing pixels indicate areas present in the original frames but absent in the model-predicted frames; the fabricated pixels show areas absent in the original frames but present in the model-predicted frames.

## 3.2 Validation of interpolation using a GCM dataset

We now apply the FILM model to a high-time-resolution dataset of Phanerozoic surface air temperature (SAT; Scotese et al., 2021). This dataset is based on GCM simulations (HadCM3L; Valdes et al., 2021), with the $CO_2$ level in the simulation inferred from global temperature proxies such as biogenic calcite and apatite $\delta^{18}O$ and lithological climate indicators. The Phanerozoic SAT dataset shares the same spatial resolution as the PaleoDEM dataset, with a resolution of 1×1 degrees, and comprises a 361×181 data array. The SAT dataset features a 10-Myr temporal resolution from 540-450 Ma and a 5-Myr resolution from

450 Ma to the present. We selected the SAT dataset from 450 Ma onward to ensure a consistent validation.

During validation, we used the SAT dataset without downscaling and conducted the same numerical validation considering temporal intervals of 10 Myrs, 20 Myrs, and 40 Myrs. Similar to the results for the PaleoDEM dataset, interpolations using a 10-Myr interval demonstrated close congruence with the actual frames, as evidenced by high values of SSIM, 2D-correlation, and PSNR, along with low values of NRMSE from 450 Ma to present (see Fig. 5; Table 2). Moreover, compared to the

260 PaleoDEM dataset, the interpolation performance of GMST across different time intervals exhibited more consistent results, as indicated by closer evaluation metrics (Table 2). This is likely because the temperature fields did not contain such sharp transitions between land and ocean.

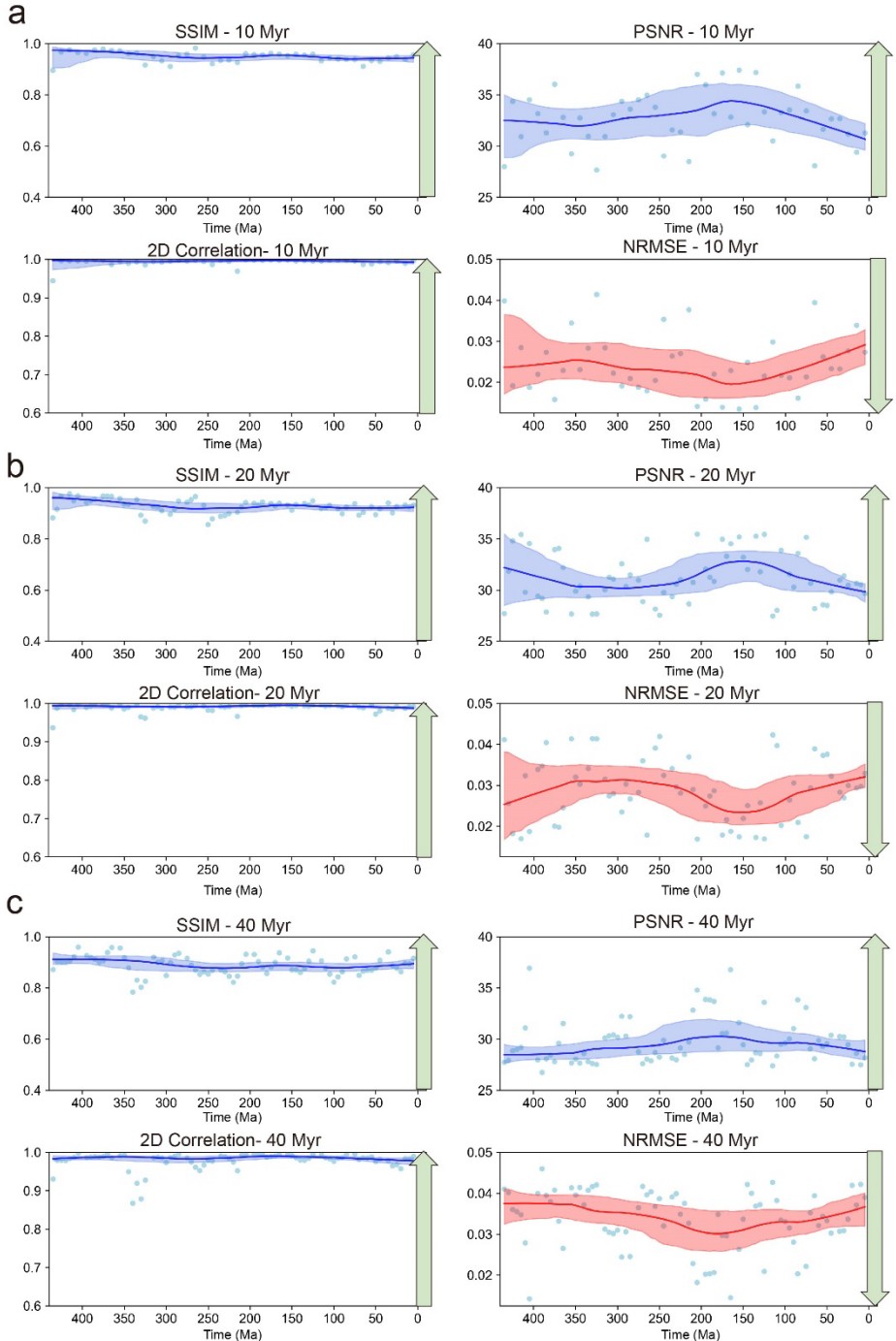

Figure 5. Comparative evaluation of performance utilizing (a) 10 Myr, (b) 20 Myr, and (c) 40 Myr intervals within the SAT Dataset. See Figure 2 for detailed explanations of the image symbols.

**Table 2. Numerical evaluation on the Phanerozoic SAT dataset**

| | 10 Ma | | | 20 Ma | | | 40 Ma | | |
|---|---|---|---|---|---|---|---|---|---|
| | Mean | Median | SD | Mean | Median | SD | Mean | Median | SD |
| SSIM | 0.95 | 0.95 | 0.02 | 0.93 | 0.93 | 0.03 | 0.89 | 0.89 | 0.04 |
| PSNR | 32.66 | 32.82 | 2.53 | 31.14 | 30.80 | 2.40 | 29.72 | 29.16 | 2.19 |
| 2D correlation | 0.99 | 0.95 | 0.02 | 0.99 | 0.99 | 0.01 | 0.98 | 0.99 | 0.02 |
| NRMSE | 0.02 | 0.02 | 0.00 | 0.03 | 0.03 | 0.00 | 0.03 | 0.03 | 0.00 |

**3.3 Output and validation of intermediate FOAM temperature and runoff datasets**

We now focus on the temperature and runoff data in our interpolated DeepFOAM dataset. Given the established correlation between increased $CO_2$ levels and a rise in global average temperature and total runoff, we anticipated that our interpolated data should mirror this trend if FILM is effectively applied to our dataset. Consequently, we test the alignment of interpolated temperature and runoff trends with those of the original model forcings. The SCION model forcing dataset is constructed with 26 distinct $CO_2$ levels, encompassing an extensive range of $CO_2$ concentrations, from 10 ppm to 112,000 ppm (Mills et al.,

2021). This dataset has been extrapolated and in-filled from an average of 5 runs of FOAM per continental configuration, which was made possible because of a predictable logarithmic response of temperature and runoff to $CO_2$ change in the model. Such a wide range of $CO_2$ levels is required to aid in model spinup where the model conditions can be far from equilibrium. Typically, Phanerozoic runs of the SCION model do not stray beyond the range of the initial dataset from FOAM. Figure 6 plots the global average temperature and runoff over these 26 $CO_2$ levels, with each of the 22 lines representing a unique time

interval (i.e., continental configuration) in FOAM. It exhibits an overall upward trend in temperature and runoff as $CO_2$ concentrations ascend, and the relationship between $CO_2$ and climate is dependent on the continental configuration (e.g. Wong Hearing et al. 2021) and solar constant. It should be briefly noted that the 0 Ma climate ensemble is computed from only one run of FOAM at preindustrial $CO_2$, adding a generalized trend for higher and lower $CO_2$ levels. This is because the SCION model is not designed to perform variable-$CO_2$ simulations at present day conditions, and only uses this state for spin up and

parameter tuning.

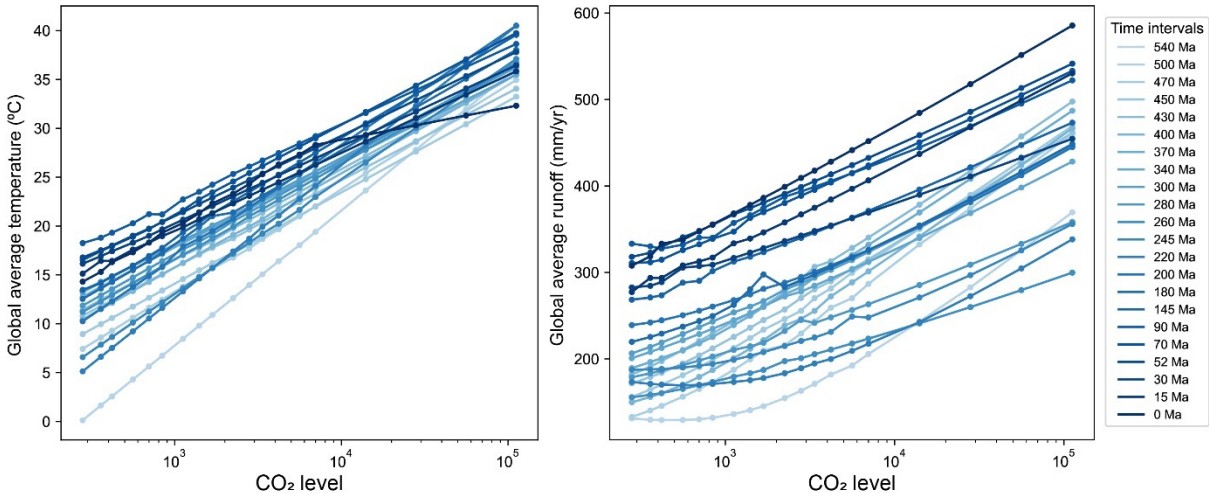

**Figure 6. Global average surface air temperature and average continental runoff over $CO_2$ levels in the FOAM dataset.** The representation is color-coded with a descending blue gradient, where light blue represents more ancient conditions, and dark blue denotes more recent scenarios. Note that the 0 Ma curve is built from only one $CO_2$ level (preindustrial), with an arbitrary increase in temperature and runoff applied when $CO_2$ levels change.

Given the known behaviour of the FOAM climate model, our FILM-interpolated temperature and runoff grids should exhibit similar patterns, and serve as a method to evaluate the effectiveness of the FILM interpolation technique for this purpose. Figure 7 plots global average temperature changes in response to varying $CO_2$ levels for each of the intermediate frames produced by FILM. The FOAM dataset contains values across 22 time intervals, leading to the 21 subplots (full subplots in Appendix) that show the average temperature changes for all intermediate frames between these 22 time intervals. The interpolated temperature values are well-aligned with the original dataset, displaying a consistent trend for escalating $CO_2$ concentrations. Figure 8 shows the runoff trends in response to varying $CO_2$ levels. Similar to the temperature trends, they exhibit patterns with respect to $CO_2$ that are analogous to those of the original runoff data. Given the more scattered distributions and more substantial changes in runoff between continental configurations, the interpolated average runoff exhibits a more varied pattern between the keyframe images, with interpolated intermediate runoff averages that can be both higher and lower than the end-members form which they are derived. Capturing runoff changes on small land patches remains a particularly challenging task for the deep learning methods, unlike the case with homogeneous variables like temperature, where FILM is more adept.

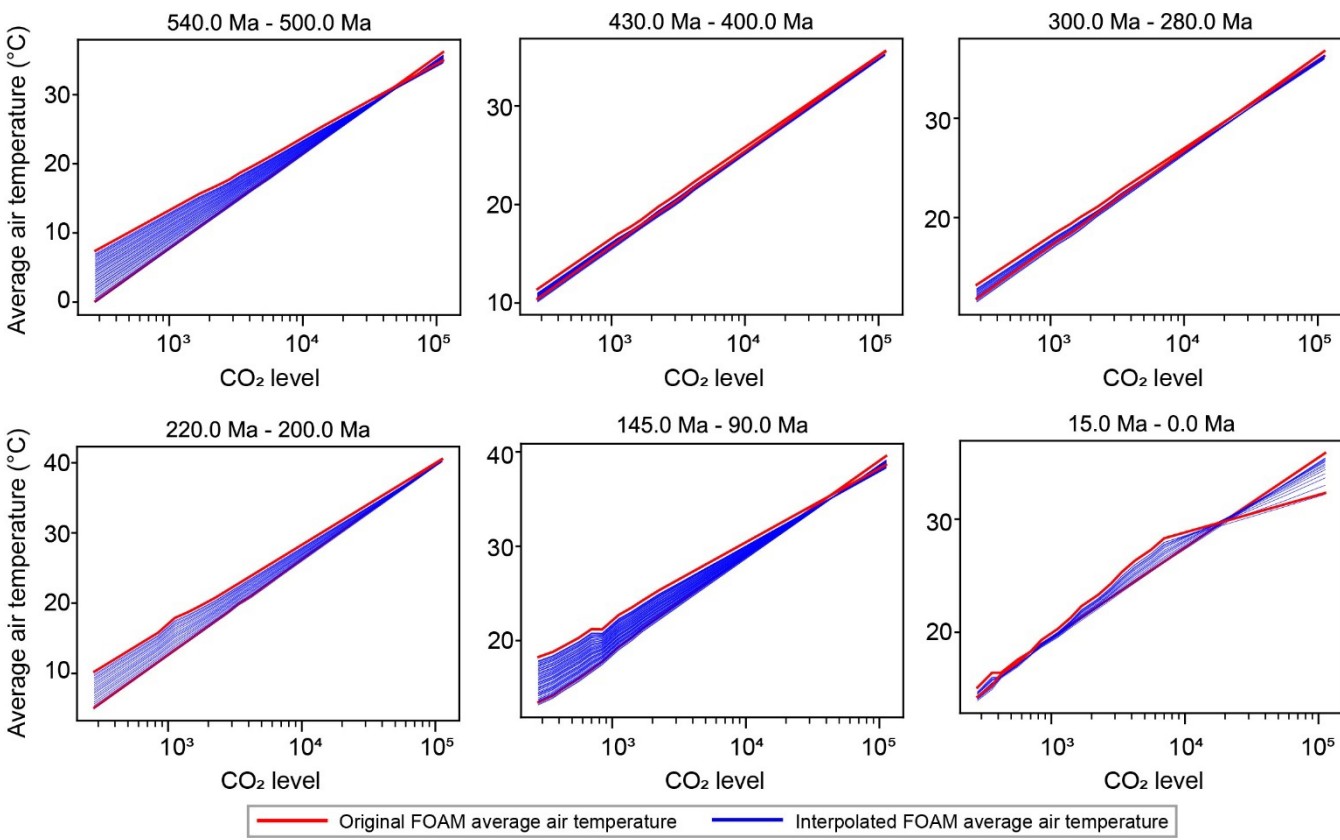

**Figure 7. Trends in global temperature changes corresponding to varying CO₂ levels.** Each subplot features one of the 21 distinct time intervals between members of the FOAM dataset. Within each subplot, the red lines delineate the keyframe average temperature variations and the blue lines show the Deep Learning-interpolated average temperature at each 1 Myr. See Fig. A1 for the full 21 subplots.

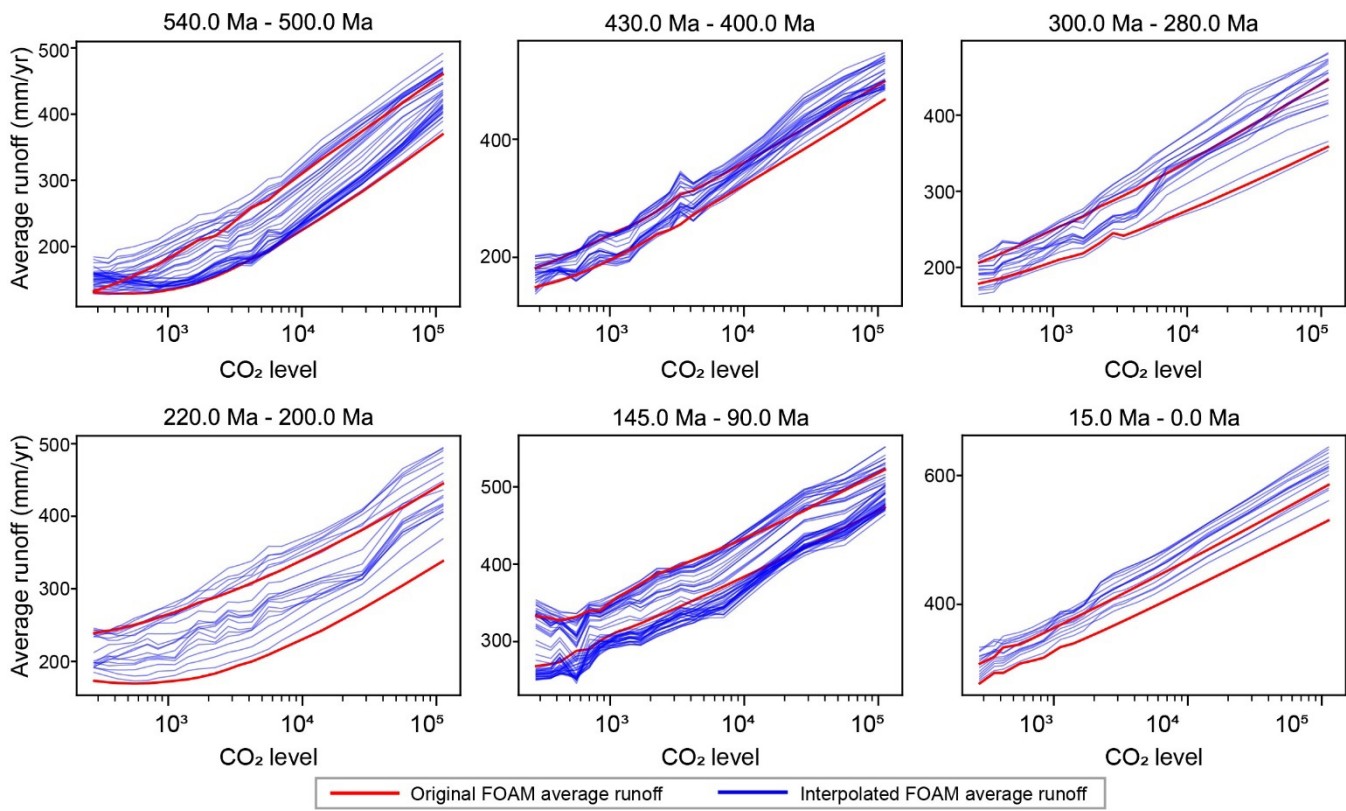

**Figure 8. Trends in runoff changes corresponding to varying CO₂ levels.** As with Fig. 7, the subplots represent the runoff changes between the original runoff outputs from the FOAM dataset. The red lines are original average runoff in the FOAM dataset, and blue lines are Deep Learning interpolated data. See Fig. A2 for the full 21 subplots.

## 4 One-million-year-forcing outputs of the SCION model and comparisons to original model.

We now run the SCION model (version 1.1.6; github.com/bjwmills/SCION) subject to the new 'DeepFOAM' dataset, which directly replaces the FOAM dataset used in the standard model. This update requires no additional modification of the SCION model. The key model predictions for atmospheric CO₂, atmospheric O₂ and global average surface temperature are shown in Fig. 9. These new model predictions follow the original model closely at the defined time points for the FOAM dataset, but they have a more detailed structure between these points and show several interesting deviations from the previous model, which are due to the new FILM-interpolated climate fields which have replaced linear interpolation between the widely-spaced previous fields.

Most notably, the SCION-DeepFOAM output for atmospheric CO₂ shows a warming spike around the Permian-Triassic boundary and a cooling spike in the Early Jurassic. These results are in line with geological evidence for extreme warmth at the Permian-Triassic extinction (Berner, 2002; Fielding et al., 2019; Yang et al., 2021; Wu et al., 2024) and a cool early Jurassic (Scotese et al., 2021). To investigate these outputs, Figure 10 plots variations in runoff and chemical weathering rates spanning 260-245Ma. The original FOAM dataset contains runoff values at 260 Ma and 245 Ma, both of which indicate high runoff in

the low latitudes (marked by grey arrows in Fig. 10) surrounding the Paleo-Tethys Ocean. As a consequence, chemical

weathering rates in these zones are comparably high, being influenced by both runoff and temperature (Maffre et al., 2018; Mills et al., 2021). Contrastingly, for the 253 Ma interval, predicted via Deep Learning, the South China plate exhibits diminished runoff values and correspondingly lower chemical weathering rates (marked by red arrows in Fig. 10). The observed decrease in weathering aligns with geological records which highlight significant aridity in China during the Permian-Triassic Boundary (Cui and Cao, 2021; Xu et al., 2023), and it is this reduced chemical weathering that leads to elevated

atmospheric $CO_2$ predictions for this time in the SCION model. However, this result requires further scrutiny, as while the Deep Learning approach affords a continuous dataset, there is no specific physical mechanisms underpinning the results. In reality, aridity here may have been due to extreme warming following the emplacement of the Siberian Traps, which is not included in our model. Moreover, variations in different paleogeographic map version (e.g., South China is smaller in Marcilly et al. 2021 than in Scotese and Wright, 2018), image processing techniques such as downscaling or upscaling, as well as the

large time intervals (>10 Myrs) between the original frames, may further complicate the results. Testing this hypothesis still requires a climate model run for the period of interest. Notably, the Deep Learning interpolation can produce intervals of climatic changes in climate-biogeochemical model, but it does not allow it to resolve climate events that were previously undetectable. For example, the Hirnantian Ice age cannot be represented in the SCION model using the DeepFOAM dataset, because various suggested mechanism for Hirnantian cooling, such as rapid weathering and a decrease in degassing dur to arc-

continent collision (Macdonald et al., 2019) and weathering amplification due to land plant evolution (Lenton et al., 2012), are not incorporated in the current SCION model used in this study (Mills et al., 2021).

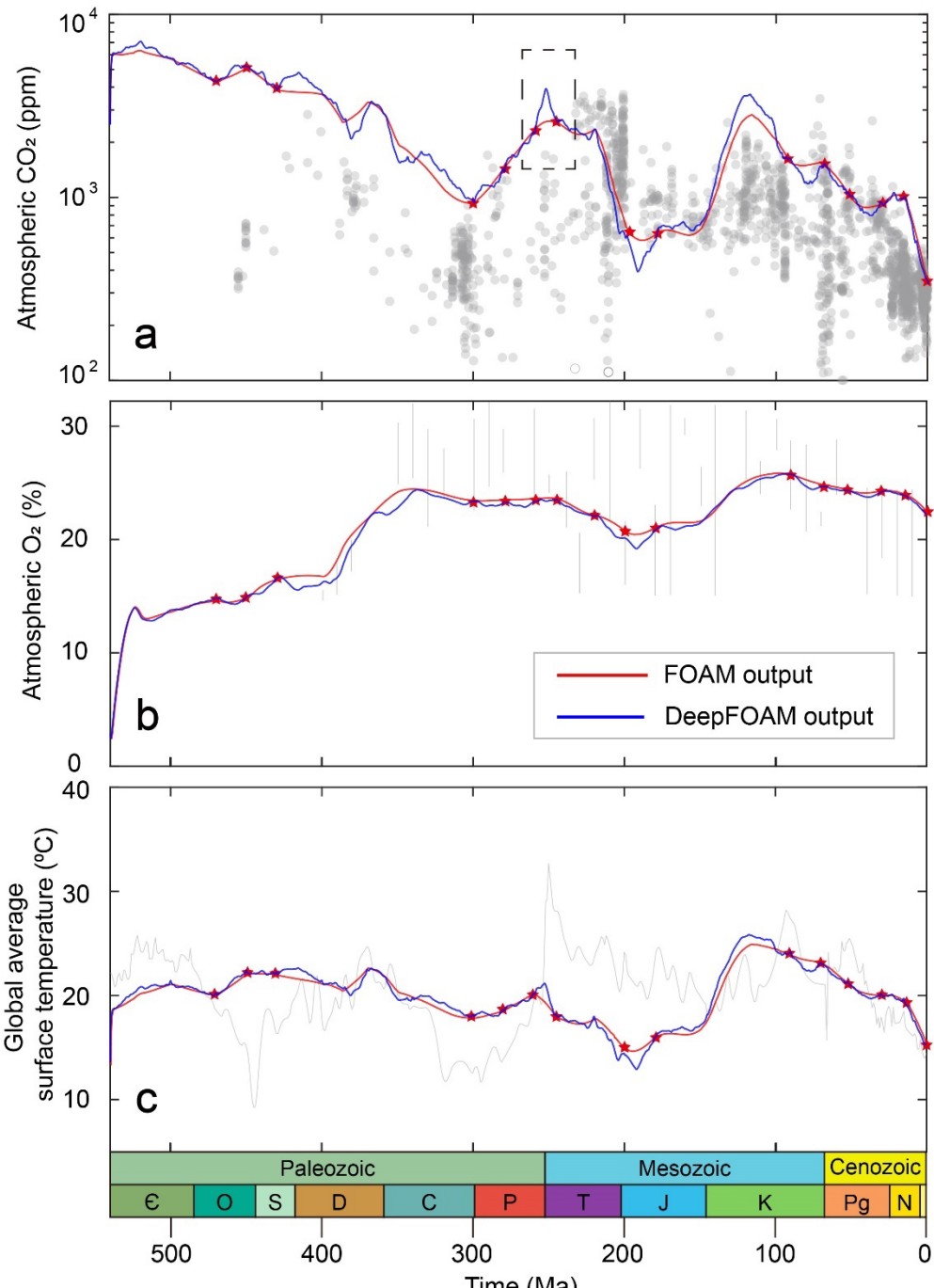

**Figure 9. Phanerozoic output comparisons between the SCION-FOAM and SCION-DeepFOAM.** (a) atmospheric $CO_2$ concentration (proxy data represented by scatter symbols; sources: Foster et al., 2017; Witkowski et al., 2018), (b) atmospheric $O_2$ concentration (proxy data represented by vertical lines; sources: Glasspool and Scott, 2010; Lenton et al., 2016), and (c) global average surface temperature (proxy data represented in gray; source: Scotese et al., 2021). The red stars on the diagram represent the time intervals of 20 Myrs or less in the FOAM dataset. The dashed box in Fig. 9a marks the significant $CO_2$ increase at 253Ma.

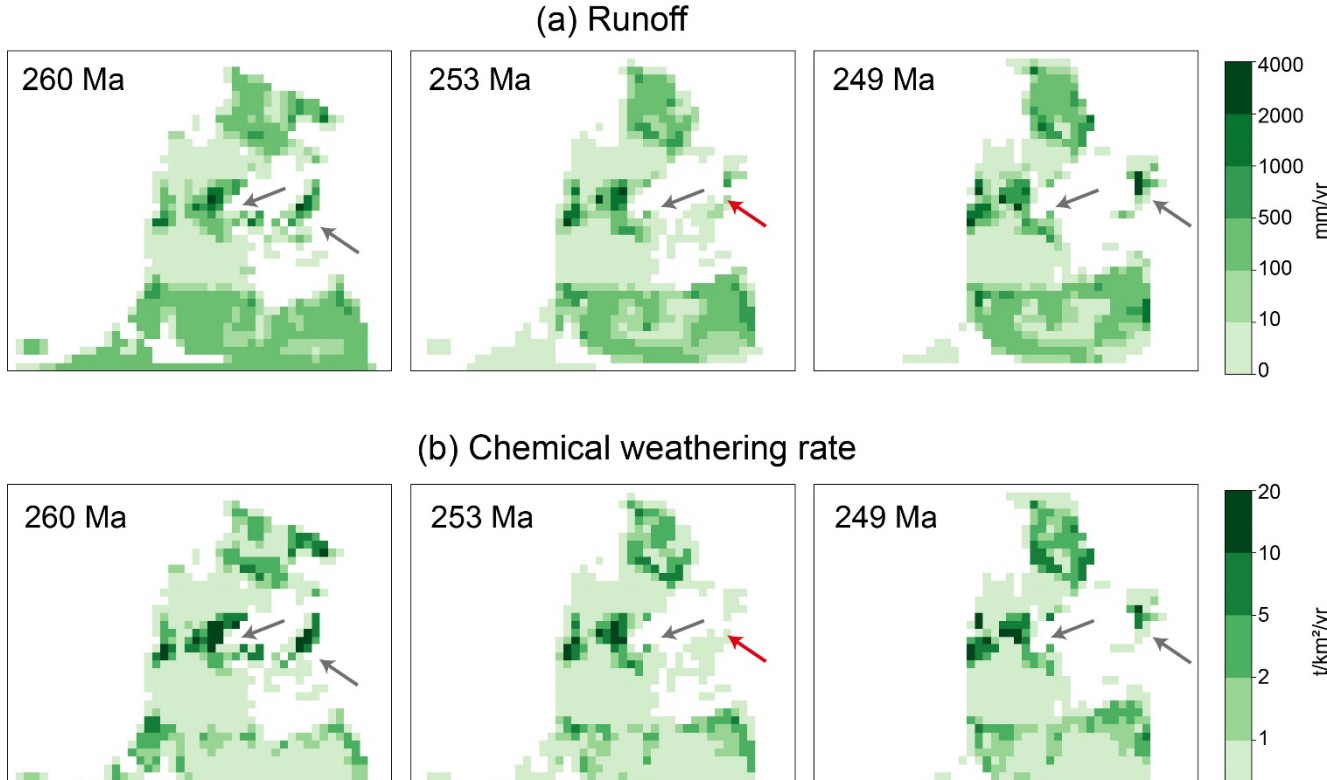

**Figure 10. Runoff (a) and chemical weathering rates (b) for 260 Ma, 253 Ma, and 249 Ma.** During the 260-245 Ma period, the low-latitude regions surrounding the Paleo-Tethys Ocean exhibit high runoff and corresponding high weathering (highlighted by grey arrows). These characteristics are derived from the FOAM dataset. However, during the 253 Ma interval, the Deep Learning method predicts reduced runoff and corresponding decreased weathering. This reduction is primarily attributed to the lowered runoff in eastern low-latitude plates, such as North and/or South China (marked by the red arrow).

## 5 Conclusions and future work

We show that deep learning can produce realistic continuous plate geographical motions, and associated paleoclimates, from snapshots up to 40 Myrs apart. The FILM Deep Learning technique can be applied to the forcing set for the SCION climate-biogeochemical model, which reduces the need for the model to interpolate linearly between time points, and thus allows a greater degree of climate variability, as well as making the model easier to use for testing specific events at known times that are not within its original forcing set. This alteration produces new intervals of climatic change in the climate-biogeochemical model, but it does not allow it to resolve any climate events that it previously could not, such as the Hirnantian ice age. It should also be noted that variations in different paleogeographic map versions, image processing techniques, and the large time intervals (>10 Myrs) and relatively coarse resolution of original frames can affect the accuracy of the interpolation. Particularly, the efficacy of interpolating runoff data, compared to that of paleogeography and temperature, is diminished by

its heterogeneous distribution. The new forcing set creates important differences in the model output, demonstrating the utility of Deep Learning for rapid preliminary analysis, but further conclusions on these differences rely on performing new climate model runs at these specific times. It is intuitively understood that these interpolation results can be enhanced with a higher-resolution original dataset, a presumption corroborated by the PaleoDEM and the SAT validation. In the PaleoDEM and the SAT validation process, interpolation results derived from a 10 Myr interval exhibited superior accuracy compared to those

from 20 Myr and 40 Myr intervals. Thus, future work to link paleoclimate and biogeochemistry should aim to run climate models at least every 10 Myrs. By combing the Deep Learning interpolation to upscale this to 1 Myr or finer time resolutions, it would allow more precise investigation of the paleoclimate and fossil record for specific events, and may also permit new approaches where modelled surface processes (e.g. vegetation) are able to distribute in space from one timestep to the next.

**Appendix A: Additional figures**

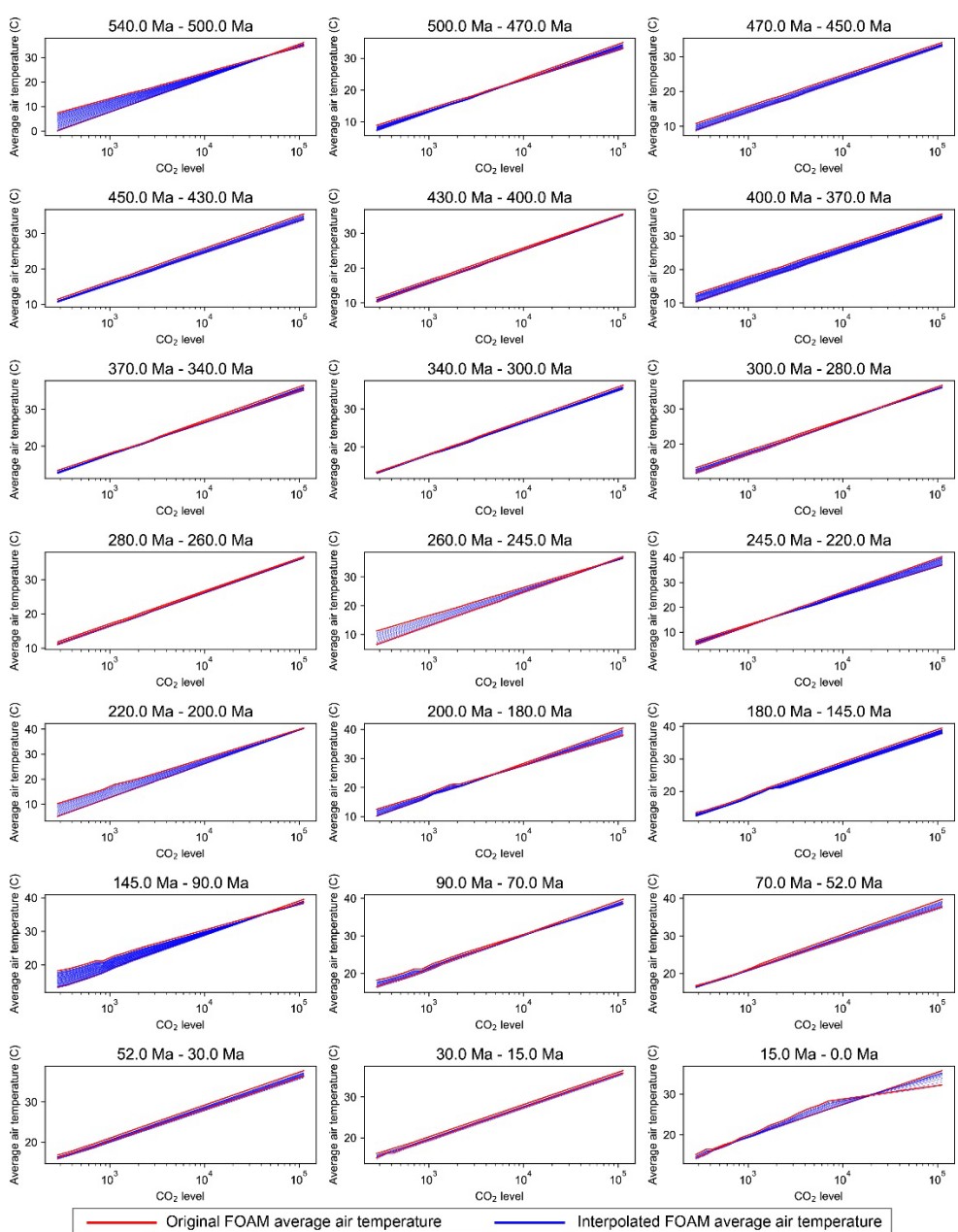

Figure A1. The full subplots of the trends in global temperature changes corresponding to varying $CO_2$ levels. Each subplot features one of the 21 distinct time intervals between members of the FOAM dataset. Within each subplot, the red lines delineate the keyframe average temperature variations and the blue lines show the Deep Learning-interpolated average temperature at each 1 Myr.

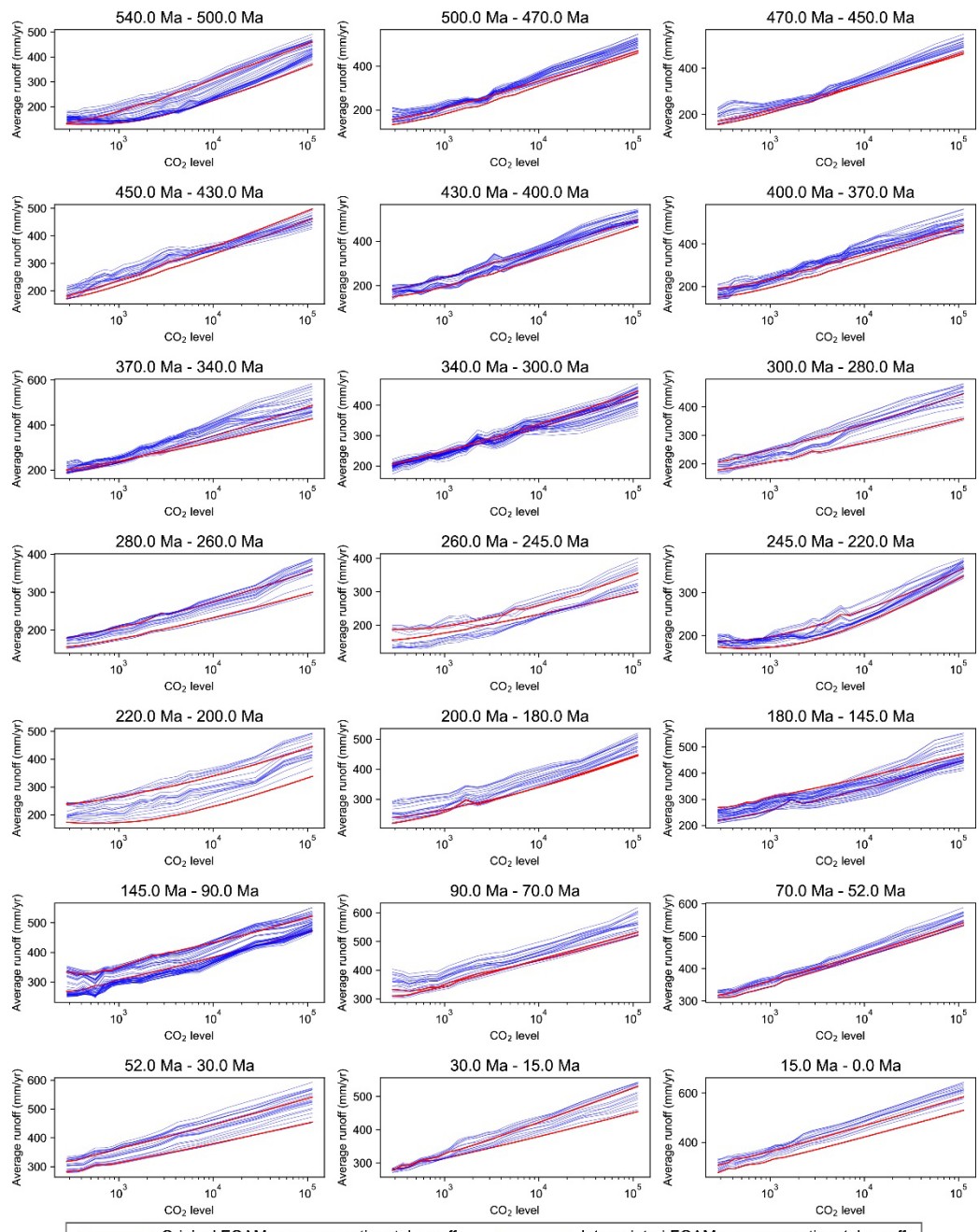

**Figure A2. The full subplots of the trends in global runoff changes corresponding to varying CO₂ levels.** Each subplot features one of the 21 distinct time intervals between members of the FOAM dataset. Within each subplot, the red lines delineate the keyframe average runoff variations and the blue lines show the Deep Learning-interpolated average runoff at each 1 Myr.

## Code availability

The FILM code is available at https://github.com/google-research/frame-interpolation and Zenodo repository (Reda et al., 2024 https://zenodo.org/records/10602810) under the Apache 2.0 licence. SCION v1.1.6 code is available at https://github.com/bjwmills/SCION and permanently archived on Zenodo (Mills, 2023 https://zenodo.org/records/7790169), codes for figure creation are archived on Zenodo (Zheng, 2024 https://zenodo.org/records/10578608).

## Author contribution

DZ, ASM, and BJWM designed this study; YG and YD provided the FOAM dataset; DZ performed the interpolation and validation, supervised by BJWM; DZ and BJWM wrote the manuscript draft; all authors reviewed and edited the manuscript. The project administration was done by BJWM.

## Competing interests

The authors declare that they have no conflict of interest.

## Acknowledgements

The authors would like to thank Stephen Hunter for useful discussions. DZ is funded by the National Science Foundation of China (No. 42202125), Natural Science Foundation of Sichuan Province (No. 24NSFSC4764), and IUGS-Deep-time Digital Earth Big Science Program. BJWM is funded by the UKRI project NE/X011208/1.

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
