# Peer review of "Using Deep Learning to integrate paleoclimate and global biogeochemistry over Phanerozoic time"

_Geoscientific Model Development, 2023_

## Referee Comment (RC1)

**Review of the article "Using Deep Learning to integrate paleoclimate and global geochemistry over Phanerozoic time."**

**General comments:**

Spatially resolved modeling has gained popularity in recent years and has proven to be a very effective tool for assessing paleoclimates. Due to the sparse data (modeled or from proxies) used in assessing fluxes through time, these models are limited. Building on SCION's already important development in this domain, this contribution develops a method that makes the model even more reliable. At least two reasons make this study very valuable and interesting:

1.    It provides a framework to improve the accuracy of surface processes used in biogeochemical models and demonstrates that 10Myrs is sufficient which gives valuable information to the community for future studies.

2.    This method reduces the global computing costs of running models over geological timescales, which are currently a major limiting factor in many research projects.

It was a pleasure to read this manuscript, which is of high quality. The scientific significance is excellent as the sparse data available is a limiting agent in the paleoclimate modeling domain. The scientific quality is very good as studies are carried out to test the reliability of this method and so on different timescales and targets (paleogeography vs runoff) and discussed in some detail. The scientific reproducibility is good as the method is explained thoroughly and models are available to download. The presentation quality is good as the figures are very relevant to the text and illustrate it well.

However, the PaleoDEM validation revealed some serious issues with interpolating over 40 Myrs, yet the authors used it on GEOCLIM/SCION with intervals up to 55 Myrs. Additionally, Figure 4 shows artificial landmasses created with a 10Myr timestep that may pose a problem for climate modeling, mostly for models including oceanic circulation. Therefore, some caution on the use of this method should be highlighted a bit more thoroughly in the text.

**Specific comments:** *(the numbering applies to the preprint version of the manuscript available online)*

| Lines | Comment |
|---|---|
| title | Phanerozoic time sounds a bit odd to me, why not use the normal word for it: "eon"? |
| 26-27 | "Species distribution", you don't mention this point in the main text, maybe it will be better placed in the conclusion as an opening? |
| 38 | Goddéris et al. (2023) is quoted but can't be found in the reference list. |
| 44 | It is mentioned 22 continental configurations. However, in both GEOCLIM and SCION, only 21 frames are cited (not assigned to any reference though). What and when is this 22$^{nd}$? |
| 47-49 | From what I understood the ITCZ is more or less forced by the model, is that what you mean? However, the ITCZ's shape and location will be determined by the paleographical configuration and be modeled by the GCM, along with the areas of extreme weathering. In most cases, it is not the other way around. Would it be possible to reformulate this sentence? |
| 60 - 63 | When I read this manuscript, what for me is the heart of this study is the method and validation parts, the latter is not mentioned at all in the introduction. |

| | |
|---|---|
| **68** | 4.5°*7.5° might be too coarse to display important paleogeographic features such as island arcs that might have a great impact on climate over some periods of the Earth's history (see Ribeiro et al., 2022 or Marcilly et al. 2022) |
| **68-69** | "Roughly evenly spaced" You then mention later in the text (l 137) that some spacing is 55 Myrs. Moreover, it is never mentioned in the text what is the model used for these continental reconstructions. If it is indeed the one used in the original GEOCLIM (Goddéris et al., 2014) then is it the maps from Blakey (2007)? (As in Nardin et al. 2011) I think this study will gain from referencing the paleogeographic models in a better way because they are at the base layer for climate modeling and therefore are extremely important. |
| **71** | "Original FOAM", which version of the model do we speak about? |
| **75** | "Wide spacing in time between …datasets": the time spacing is mentioned but still no numbers are given → maybe it will be nice to give some numbers for the reader to know which scale of spacing we are talking about? |
| **95-96** | You mention shifts in FOAM are due to the reorganization of landmasses, yet no real plate tectonic/paleogeography studies are quoted here. It will be nice to have the study presenting the reconstructions behind FOAM quoted. |
| **142** | Maybe "the" should be "a" PaleoDEM dataset. PaleoDEM is a quite widely used term and not restricted to the work of Scotese. |
| **154** | The problem with downscaling is that it often results in an overestimation of the exposed land ratio which will mean more area available for weathering for the biogeochemical model. I know it's difficult to run GCM with a finer grid.
 *(It was just to raise awareness as this is not strictly related to the subject of your study. I understand the point here is to demonstrate the reliability of the method which I think is very well done in this study)* |
| **227-228** | I'm not sure what you are trying to say here: that the synthetic and real maps (Scotese and Wright, 2018) have a better fit together compared to Scotese& wright (2018) and Marcilly et al. (2021)? If so, it might need some reformulation.
 What period of comparison are we talking about here?
 Can you give an estimate of this discrepancy, in % error for example? |
| **274 - 275** | Having issues with small landmasses is quite serious because they often display high runoff (Goddéris et al., 2014) and therefore host high weathering. |
| **294-296+ § 4** | This is interesting because other models such as GEOCARBSULF (Marcilly et al., 2021) and GEOCLIM (Goddéris & Donnadieu, 2017; Goddéris et al, 2014) have this spike which is attributed to a change in climate sensitivity in GEOCARBSULF for example.

 I'm confused about how the frames are now interpolated; in this section, are we back to the first part where you interpolate following the spacing of the maps which are roughly evenly spaced"? how much time in between two frames? It is a bit confusing after the validation part with the 3 different spacings.

 So, you demonstrate that the accuracy with spacing greater than 10Myrs is reduced using this method and yet you use it with intervals up to 55 Myrs? Is that not a problem? I don't think you should draw any conclusions with intervals over 10 Myrs which if it is indeed the FOAM runs as the one in Goddéris et al. (2014), actually covers the majority of the Phanerozoic. |

[Figure]

Time spacing between 2 maps from Goddéris et al. (2014)

Age (Ma)

| | This is where it becomes complicated for me to understand because the extreme warmth of the Permian Triassic extinction is probably shorter than 10 Myrs so can you actually see this signal? in Cao et al (2022) the interval considered is 253-247 Ma, roughly 254-250 Ma in Yang et al. (2019) for example.

How can you see short signals such as the P/T warming but not the Ordovician cooling (Hirnantian) for example? (Which has been attributed by many to be caused by changes in paleogeography (e.g., Nardin et al., 2011).

Why quoting Wu et al. (2023) the article in the reference list is not about the PT boundary.? |
|---|---|
| **301** | The fate of South China will also depend on the chosen reconstruction and downscaling process as for the lower Triassic in Fig 9 South China seems well emerged but in the reconstructions of Marcilly et al. (2021) the land area available for weathering is very small. |
| **305 - 307** | You should also mention that the timestep between two "base" reconstructions is greater than 10 Myrs and the accuracy is therefore reduced. |
| **323** | "20 Myrs apart" are they though? From Goddéris et al. (2014) they seem more 30 to 40 Myrs apart for the majority.

"Can be applied "vs "should aim to run climate models at least every 10 Myrs" (**l333**) Therefore, the recommendation made for further studies is not respected in this very study → a bit of mixed message. Maybe this sentence should be rewritten? |
| **334** | The conclusion is very well structured and easy to read. Maybe it will also be worth mentioning here that the method fails to reconstruct short-lived events (greater than 1Myrs though) such as the Hirnantian glacial event. Even though they have large climatic consequences. |

Comment related to figures.

| **Figure** | **Comment** |
|---|---|
| **2** | Concerning the graphs presenting SSIM and 2D correlation:
Whatever the frame interval considered it seems that there are two periods of increasingly low performance. It's difficult to read for sure the ages but I would say between 430-420 Ma and 250-210 Ma. What can cause this? |

[Figure]

| **Table 1** | Title: Missing the a in "evaluation" |
|---|---|
| **4** | The synthetic maps at 105 Ma (yellow arrows) worry me a bit because it's ok (not ideal though) for running GCM simulations with FOAM because it doesn't have a proper oceanic circulation module but with other GCMs such artifact will represent an issue. Can you comment on that?

You don't mention it in the text but this creation of land over South China (orange arrow) will create a big issue for the assessment of weathering fluxes at it is well known small, isolated landmasses are hosting a lot of runoff and therefore weathering (mostly at the equator). It will be nice to highlight this point even if you already mentioned that the 40 Myrs step is less accurate. It will actually illustrate this point.

[Figure]
 |
| **5** | In both the runoff and temperature graph, deeper time runs seem to have a better correlation between $CO_2$-Temprature and $CO_2$-runoff: How do you explain that? |
| **6-7** | Those two figures are quite crowded, is it possible to select the most representative graphs and put the other ones in the appendix? |

| | |
|---|---|
| 8 | It will be nice to see an estimation of the "accuracy" of the method on this figure. Maybe highlight the periods where base maps are closer to each other and so lead to more accuracy. This way the reader can directly see which periods are more reliable than the others.

In the text you mention, that with this method, the timestep is reduced to 1Myr so we should see the signal of more short-lived events. However, here the Hirnantian cooling is totally hidden and instead there is even an increase in $CO_2$ and temperature. Can you comment on this? |
| 9 | The figure highlights the increase in runoff and weathering in central Pangea during the Late Permian /Early Triassic in mid-latitude Pangea, but this is debated and evidences such as large extend of evaporites deposits suggest quite arid conditions (Scotese maps below (DOI:10.13140/2.1.2757.8567.) or Cui & Cao (2021) https://doi.org/10.1002/gj.4123).

 Arid conditions are unlikely to result in intense weathering. Can you comment on that as well?

[Figure]
 |

---

## Author Comment (AC1)

**Author responses to Review comments for:**

**Using Deep Learning to integrate paleoclimate and global geochemistry over Phanerozoic time**

Dongyu Zheng, Andrew Merdith, Yves Goddéris, Yannick Donnadieu, Khushboo Gurung, and Benjamin J. W. Mills

We thank the reviewers for their insightful comments and have addressed all the points raised. We have incorporated several important considerations into the discussion and conclusion sections for readers who may wish to use their own datasets for interpolation. Additionally, we conducted a numerical evaluation using a high-resolution temperature dataset based on GCM simulations. This evaluation further demonstrates that the Deep Learning method can indeed produce reliable interpolations of climatic variables.

(Black text = reviewers' comments; Red text = our response; Blue text = additions/changes to paper, line numbers refer to the clear revised manuscript)

**Response to Reviewer 1**

**General comments:**

Spatially resolved modeling has gained popularity in recent years and has proven to be a very effective tool for assessing paleoclimates. Due to the sparse data (modeled or from proxies) used in assessing fluxes through time, these models are limited. Building on SCION's already important development in this domain, this contribution develops a method that makes the model even more reliable. At least two reasons make this study very valuable and interesting:

1. It provides a framework to improve the accuracy of surface processes used in biogeochemical models and demonstrates that 10Myrs is sufficient which gives valuable information to the community for future studies.

2. This method reduces the global computing costs of running models over geological timescales, which are currently a major limiting factor in many research projects.

It was a pleasure to read this manuscript, which is of high quality. The scientific significance is excellent as the sparse data available is a limiting agent in the paleoclimate modeling domain. The scientific quality is very good as studies are carried out to test the reliability of this method and so on different timescales and targets (paleogeography vs runoff) and discussed in some detail. The scientific reproducibility is good as the method is explained thoroughly and models are available to download. The presentation quality is good as the figures are very relevant to the text and illustrate it well.

However, the PaleoDEM validation revealed some serious issues with interpolating over 40 Myrs, yet the authors used it on GEOCLIM/SCION with intervals up to 55 Myrs. Additionally, Figure 4 shows artificial landmasses created with a 10Myr timestep that may pose a problem for climate modeling, mostly for models including oceanic circulation. Therefore, some caution on the use of this method should be highlighted a bit more thoroughly in the text.

We agree with reviewer's comments and thank them for the positive assessment. Indeed, although the FILM method can provide potential continuous interpolated frames that can be used for models like SCION, caution is absolutely warranted if using these in the place of GCM simulations to make conclusions. This is especially true when considering large time spacings (>10 Myrs) which will complicate the interpolation results. We have incorporated these considerations in the revised manuscript and have been careful to point this out in the abstract, modifying the final phrase to "Consequently, interpolated climates must be confirmed by running a paleoclimate model if scientific conclusions are to be based directly on them."

Lines 30-31: … heterogeneous distribution of runoff. Consequently, interpolated climates must be confirmed by running a paleoclimate model if scientific conclusions are to be based directly on them.

**Specific comments:** *(the numbering applies to the preprint version of the manuscript available online)*

**Lines Comment**

| | |
|---|---|
| **title** | Phanerozoic time sounds a bit odd to me, why not use the normal word for it: "eon"?

 Modified. |
| **26-27** | "Species distribution", you don't mention this point in the main text, maybe it will be better placed in the conclusion as an opening?

 Deleted. |
| **38** | Goddéris et al. (2023) is quoted but can't be found in the reference list.

 Added. |
| **44** | It is mentioned 22 continental configurations. However, in both GEOCLIM and SCION, only 21 frames are cited (not assigned to any reference though). What and when is this 22$^{nd}$?

 For the FOAM dataset, I am using the FOAM dataset in SCION v1.1.6 at https://github.com/bjwmills/SCION. It has a total of 22 time intervals at 540, 500, 470, 450, 430, 400, 370, 340, 300, 280, 260, 245, 220, 200, 180, 145, 90, 70, 52, 30, 15, 0 Ma. The original FOAM runs were 21 paleogeographic configurations and one map for present day, so 22 in total. The original SCION model v1.0.0 only used 21 simulations, omitting a frame at 15 Ma where the topographic data was difficult to reconcile with the climate – which was later resolved in v1.1.0. We have added more detail here and these time intervals are now presented in the legend of Figure 6. |

[Figure]

Figure 6. Global average surface air temperature and average continental runoff over CO2 levels in the FOAM dataset.

| 47-49 | From what I understood the ITCZ is more or less forced by the model, is that what you mean? |
| | However, the ITCZ's shape and location will be determined by the paleographical configuration and be modeled by the GCM, along with the areas of extreme weathering. In most cases, it is not the other way around. Would it be possible to reformulate this sentence? |
| | We have reworded this to describe continents in the humid low latitudes, rather than focusing on the ITCZ specifically. |
| | Lines 47-50: For example, through plate tectonic motion, a mountain range may pass through the tropics, an event expected to cause a spike in continental weathering due to high rainfall, but this may be undetected by SCION or GEOCLIM if the timespan at which the mountain range crossed the equator was not represented in the time points chosen for the paleoclimate simulations. |
| 60 - 63 | When I read this manuscript, what for me is the heart of this study is the method and validation parts, the latter is not mentioned at all in the introduction. |
| | We have added an introduction of these validations in the Introduction Section. |
| | Lines 59-62: In this paper, we first performed a numerical and visual validation of the Deep Learning interpolation of a PaleoDEM topographic elevation dataset (Scotese and Wright, 2018), as well as surface air temperature generated from these maps using the HadCM3L GCM (Scotese et al., 2021; Valdes et al., |

| | |
|---|---|
| | 2021). The validation results suggest that the Deep Learning method is capable of adequately detecting plate motions and changes to surface air temperature. |
| 68 | 4.5°\*7.5° might be too coarse to display important paleogeographic features such as island arcs that might have a great impact on climate over some periods of the Earth's history (see Ribeiro et al., 2022 or Marcilly et al. 2022)

We agree. We plan to use higher resolution paleogeographic maps for future SCION simulations. We have added a note on this in the conclusions.

Line 368: …relatively coarse resolution of original frames can affect the accuracy of the interpolation. |
| 68-69 | "Roughly evenly spaced" You then mention later in the text (l 137) that some spacing is 55 Myrs. Moreover, it is never mentioned in the text what is the model used for these continental reconstructions. If it is indeed the one used in the original GEOCLIM (Goddéris et al., 2014) then is it the maps from Blakey (2007)? (As in Nardin et al. 2011) I think this study will gain from referencing the paleogeographic models in a better way because they are at the base layer for climate modeling and therefore are extremely important.

Yes, these are the FOAM model runs from GEOCLIM as in Godderis et al. 2014 and use same paleogeographies as in that work, which are assembled from works by Blakey, Besse and Fluteau, and Sewell. We have now noted this.

Lines 69-71: The SCION model employs a series of 2D model forcing fields taken from annual means of the FOAM climate model, which were initially developed for the GEOCLIM model (Godderis et al., 2014). These fields are paleogeography (a composite of works by Blakey, Besse and Fluteau, and Sewall – see Godderis et al., 2014 for details), … |
| 71 | "Original FOAM", which version of the model do we speak about?

We wanted to highlight the difference between the FOAM model runs and the Deep Learning interpolated FOAM, and therefore use 'Original' here. We have deleted the 'Original' to avoid confusion. |

| | |
|---|---|
| **75** | "Wide spacing in time between …datasets": the time spacing is mentioned but still no numbers are given à maybe it will be nice to give some numbers for the reader to know which scale of spacing we are talking about?

We've added a simple explanation in the text, the time intervals are available in Figure 6.

Line 73: …time intervals shown in Fig. 6

[Figure]

Figure 6. Global average surface air temperature and average continental runoff over CO2 levels in the FOAM dataset. |
| **95-96** | You mention shifts in FOAM are due to the reorganization of landmasses, yet no real plate tectonic/paleogeography studies are quoted here. It will be nice to have the study presenting the reconstructions behind FOAM quoted.

We now direct the reader to Godderis et al. 2014 for the paleogeographic information.

Lines 69-71: The SCION model employs a series of 2D model forcing fields taken from annual means of the FOAM climate model, which were initially developed for the GEOCLIM model (Godderis et al., 2014). These fields are paleogeography (a composite of works by Blakey, Besse and Fluteau, and Sewall – see Godderis et al., 2014 for details), … |
| **142** | Maybe "the" should be "a" PaleoDEM dataset. PaleoDEM is a quite widely used term and not restricted to the work of Scotese.

Modified. |

| | |
|---|---|
| 154 | The problem with downscaling is that it often results in an overestimation of the exposed land ratio which will mean more area available for weathering for the biogeochemical model. I know it's difficult to run GCM with a finer grid. (It was just to raise awareness as this is not strictly related to the subject of your study. I understand the point here is to demonstrate the reliability of the method which I think is very well done in this study)

 We agree. |
| 227-228 | I'm not sure what you are trying to say here: that the synthetic and real maps (Scotese and Wright, 2018) have a better fit together compared to Scotese& wright (2018) and Marcilly et al. (2021)? If so, it might need some reformulation.

 What period of comparison are we talking about here?

 Can you give an estimate of this discrepancy, in % error for example?

 We want to emphasize that the predicted paleogeographic maps are visually similar to the original maps across points in Phanerozoic time. We have rephrased this sentence to avoid confusion and have pointed the reader to the metrics we use to quantify the discrepancies.

 Lines 233-234: Across different time periods, predicted frames were generally visually comparable to original ones (see Fig. 2 for numerical estimations). |
| 274 - 275 | Having issues with small landmasses is quite serious because they often display high runoff (Goddéris et al., 2014) and therefore host high weathering.

 We agree. We have added a note on this.

 Lines 239-241: Nevertheless, the FILM method creates a significant number of unmatched pixels compared to the original frames, which would alter climatic outputs of GCMs and linked biogeochemical calculations, especially as small introduced islands would be expected to have high runoff and chemical weathering rates (Park et al., 2020). |
| 294-296+ | This is interesting because other models such as GEOCARBSULF (Marcilly et al., 2021) and GEOCLIM (Goddéris & Donnadieu, 2017; Goddéris et al, 2014) |

| | |
|---|---|
| **§ 4** | have this spike which is attributed to a change in climate sensitivity in GEOCARBSULF for example.

I'm confused about how the frames are now interpolated; in this section, are we back to the first part where you interpolate following the spacing of the maps which are roughly evenly spaced"?  how much time in between two frames? It is a bit confusing after the validation part with the 3 different spacings.
So, you demonstrate that the accuracy with spacing greater than 10Myrs is reduced using this method and yet you use it with intervals up to 55 Myrs? Is that not a problem? I don't think you should draw any conclusions with intervals over 10 Myrs which if it is indeed the FOAM runs as the one in Goddéris et al. (2014), actually covers the majority of the Phanerozoic.

[Figure]

Correct, we determine the optimum spacing for interpolation to be about 10 Myrs but we are nevertheless using the approach on the FOAM dataset which has larger spacings, because this is the only dataset readily available for this task. We have emphasized further in the revised paper that conclusions should not be directly drawn from interpolated climates without verifying them with a GCM. See abstract and conclusions. |

Lines 30-31: …Consequently, interpolated climates must be confirmed by running a paleoclimate model if scientific conclusions are to be based directly on them.

Lines 375-377: …Thus, future work to link paleoclimate and biogeochemistry should aim to run climate models at least every 10 Myrs. By combing the Deep Learning interpolation to upscale this to 1 Myr or finer time resolutions, it would allow more precise investigation of the paleoclimate and fossil record for specific events

This is where it becomes complicated for me to understand because the extreme warmth of the Permian Triassic extinction is probably shorter than 10 Myrs so can you actually see this signal? in Cao et al (2022) the interval considered is 253-247 Ma, roughly 254-250 Ma in Yang et al. (2019) for example.

We have clarified that we do not see the Permian-Triassic extinction, which was driven by a large CO2 input. What we do see is a potential increase in background CO2 levels over a longer timeframe.

Lines 336-338: In reality, aridity here may have been due to extreme warming following the emplacement of the Siberian Traps, which is not included in our model.

How can you see short signals such as the P/T warming but not the Ordovician cooling (Hirnantian) for example? (Which has been attributed by many to be caused by changes in paleogeography (e.g., Nardin et al., 2011).

There may be many reasons why we do not see Hirnantian cooling in this model. Various suggested mechanisms for Hirnantian cooling, such as rapid weathering and a decrease in degassing due to arc-continent collision (Macdonald et al., 2019) and weathering amplification due to land plant evolution (Lenton et al., 2012), are not incorporated in the current SCION model used in this study. This limitation is also discussed in Mills et al. (2021).

| | |
|---|---|
| | Lines 341-346: Notably, the Deep Learning interpolation can produce intervals of climatic changes in climate-biogeochemical model, but it does not allow it to resolve climate events that were previously undetectable. For example, the Hirnantian Ice age cannot be represented in the SCION model using the DeepFOAM dataset, because various suggested mechanism for Hirnantian cooling, such as rapid weathering and a decrease in degassing dur to arc-continent collision (Macdonald et al., 2019) and weathering amplification due to land plant evolution (Lenton et al., 2012), are not incorporated in the current SCION model used in this study (Mills et al., 2021).

Why quoting Wu et al. (2023) the article in the reference list is not about the PT boundary.?

Thanks for pointing out the missing reference. We have corrected this to cite Wu et al. (2024) – 'The terrestrial end-Permian mass extinction in the paleotropics postdates the marine extinction'. This reference has now been added in the list. |
| 301 | The fate of South China will also depend on the chosen reconstruction and downscaling process as for the lower Triassic in Fig 9 South China seems well emerged but in the reconstructions of Marcilly et al. (2021) the land area available for weathering is very small.

We have incorporated this caution into the manuscript.

Lines 336-340: In reality, aridity here may have been due to extreme warming following the emplacement of the Siberian Traps, which is not included in our model. Moreover, variations in different paleogeographic map version (e.g., South China is smaller in Marcilly et al. 2021 than in Scotese and Wright, 2018), image processing techniques such as downscaling or upscaling, as well as the large time intervals (>10 Myrs) between the original frames, may further complicate the results. |

| | |
|---|---|
| **305 - 307** | You should also mention that the timestep between two "base" reconstructions is greater than 10 Myrs and the accuracy is therefore reduced.

Same as above. We have incorporated this caution in the manuscript.

Lines 368-369: …and the large time intervals (>10 Myrs) and relatively coarse resolution of original frames can affect the accuracy of the interpolation |
| **323** | "20 Myrs apart" are they though? From Goddéris et al. (2014) they seem more 30 to 40 Myrs apart for the majority.

Indeed, the <20-Myr interval is the most common, accounting for 52% of the total time intervals. Time intervals over 30 Myrs account for only 19% of the total. Please refer to the diagram below for further details.

[Figure]

[Figure]

"Can be applied "vs "should aim to run climate models at least every 10 Myrs" (**l333**) |

| | |
|---|---|
| | Therefore, the recommendation made for further studies is not respected in this very study à a bit of mixed message. Maybe this sentence should be rewritten?

Modified as suggested.

Lines 375-377: Thus, future work to link paleoclimate and biogeochemistry should aim to run climate models at least every 10 Myrs. By combing the Deep Learning interpolation to upscale this to 1 Myr or finer time resolutions, it would allow more precise investigation of the paleoclimate and fossil record for specific events, … |
| **334** | The conclusion is very well structured and easy to read. Maybe it will also be worth mentioning here that the method fails to reconstruct short-lived events (greater than 1Myrs though) such as the Hirnantian glacial event. Even though they have large climatic consequences.

As suggested, we note in the conclusions that SCION still cannot match the Hirnantian Ice Age.

Lines 365-366: This alteration produces new intervals of climatic change in the climate-biogeochemical model, but it does not allow it to resolve any climate events that it previously could not, such as the Hirnantian ice age. |

Comment related to figures:

| Figure | Comment |
|---|---|
| **2** | Concerning the graphs presenting SSIM and 2D correlation:

Whatever the frame interval considered it seems that there are two periods of increasingly low performance. It's difficult to read for sure the ages but I would say between 430-420 Ma and 250-210 Ma. What can cause this? |

[Figure]

Thank you for the detailed observations. Indeed, the reduced performance is observed around the intervals of ~430-420 Ma and 250-210 Ma. The SSIM and 2D correlation metrics measure the differences of luminance, contrast, and structural information between two images. We expect that the (downscaled) paleogeography at these times, and associated motions between timeframes are more sensitive to these metrics.

Lines 229-231: Interestingly, the SSIM and 2D-correlaiton show a particular decrease in performance around 220 Ma and 420 Ma. This may be due to more complex plate movements around these times which the algorithm finds more difficult to predict.

| | |
|---|---|
| **Table 1** | Title: Missing the a in "evaluation"
 Modified. |
| **4** | The synthetic maps at 105 Ma (yellow arrows) worry me a bit because it's ok (not ideal though) for running GCM simulations with FOAM because it doesn't have a proper oceanic circulation module but with other GCMs such artifact will represent an issue. Can you comment on that?

 You don't mention it in the text but this creation of land over South China (orange arrow) will create a big issue for the assessment of weathering fluxes at it is well known small, isolated landmasses are hosting a lot of runoff and therefore weathering (mostly at the equator). It will be nice to highlight this |

point even if you already mentioned that the 40 Myrs step is less accurate. It will actually illustrate this point.

[Figure]

These are good points. We have incorporated these considerations into the revised manuscript.

Lines 239-241: Nevertheless, the FILM method creates a significant number of unmatched pixels compared to the original frames, which would alter climatic outputs of GCMs and linked biogeochemical calculations, especially as small introduced islands would be expected to have high runoff and chemical weathering rates (Park et al., 2020).

| 5 | In both the runoff and temperature graph, deeper time runs seem to have a better correlation between $CO_2$-Temprature and $CO_2$-runoff: How do you explain that? |
| | Both of these effects are probably linked to the large polar supercontinent in the early Paleozoic in these reconstructions. As we do not have the full climate |

model outputs, we cannot comment on this in detail but other work with FOAM has shown that the continental configuration can alter the $CO_2$-temperature relationship significantly (Wong Hearing et al., 2021). Lines 281-282.

Lines 281-282: …and the relationship between $CO_2$ and climate is dependent on the continental configuration (e.g. Wong Hearing et al. 2021) and solar constant.

Those two figures are quite crowded, is it possible to select the most representative graphs and put the other ones in the appendix?

We have improved the figure as suggested. See revised Figures 7&8.

[Figure]

Figure 7. Trends in global temperature changes corresponding to varying $CO_2$ levels. Each subplot features one of the 21 distinct time intervals between members of the FOAM dataset. Within each subplot, the red lines delineate the keyframe average temperature variations and the blue lines show the Deep Learning-interpolated average

[Figure]

temperature at each 1 Myr. See Fig. A1 for the full 21 subplots.

Figure 8. Trends in runoff changes corresponding to varying CO2 levels. As with Fig. 7, the subplots represent the runoff changes between the original runoff outputs from the FOAM dataset. The red lines are original average runoff in the FOAM dataset, and blue lines are Deep Learning interpolated data. See Fig. A2 for the full 21 subplots.

| | |
|---|---|
| 8 | It will be nice to see an estimation of the "accuracy" of the method on this figure. Maybe highlight the periods where base maps are closer to each other and so lead to more accuracy. This way the reader can directly see which periods are more reliable than the others.

We have highlighted the time points in Figure 9 where the time intervals are less than or equal to 20 Myrs. |

[Figure]

Figure 9. Phanerozoic output comparisons between the SCION-FOAM and SCION-DeepFOAM. (a) atmospheric CO2 concentration (proxy data represented by scatter symbols; sources: Foster et al., 2017; Witkowski et al., 2018), (b) atmospheric O2 concentration (proxy data represented by vertical lines; sources: Glasspool and Scott, 2010; Lenton et al., 2016), and (c) global average surface temperature (proxy data represented in gray; source: Scotese et al., 2021). The red stars on the diagram represent the time intervals of 20 Myrs or less in the FOAM dataset. The dashed box in Fig. 9a marks the significant CO2 increase at 253Ma.

In the text you mention, that with this method, the timestep is reduced to 1Myr so we should see the signal of more short-lived events. However, here the Hirnantian cooling is totally hidden and instead there is even an increase in $CO_2$ and temperature. Can you comment on this?

As above, we have now added a comment on why the model does not reproduce the Hirnantian glaciation.

| | |
|---|---|
| | Lines 341-346: Notably, the Deep Learning interpolation can produce intervals of climatic changes in climate-biogeochemical model, but it does not allow it to resolve climate events that were previously undetectable. For example, the Hirnantian Ice age cannot be represented in the SCION model using the DeepFOAM dataset, because various suggested mechanism for Hirnantian cooling, such as rapid weathering and a decrease in degassing dur to arc-continent collision (Macdonald et al., 2019) and weathering amplification due to land plant evolution (Lenton et al., 2012), are not incorporated in the current SCION model used in this study (Mills et al., 2021). |
| 9 | The figure highlights the increase in runoff and weathering in central Pangea during the Late Permian /Early Triassic in mid-latitude Pangea, but this is debated and evidences such as large extend of evaporites deposits suggest quite arid conditions (Scotese maps below (DOI:10.13140/2.1.2757.8567.) or Cui & Cao (2021) https://doi.org/10.1002/gj.4123). Arid conditions are unlikely to result in intense weathering.  Can you comment on that as well? |

[Figure]

The relatively wet central Pangaea is an output of the FOAM GCM, so our analysis does not alter it. We have noted the Cui and Cao paper in the revision in the section about aridity.

Lines 336-337: In reality, aridity here may have been due to extreme warming following the emplacement of the Siberian Traps, which is not included in our model.

**Response to Reviewer 2**

Summary: The study presents an application of an interpolation algorithm, originally developed to interpolate video frames, to deep-time paleoclimate simulations. Using this application, steady-state snap-shot climate simulations widely separated in time can be interpolated to produce paleoclimate maps with a higher time resolution.

The manuscript uses interpolated maps of palaeogeography, temperature, and runoff covering the last 500 million years with a time resolution of 1 million years (from an original time resolution of about 25 million years) to drive a biochemical climate model.

Recommendation: I liked the study's original idea, but I am not totally convinced that the authors have demonstrated that the method can work properly. My background is in climate dynamics, statistical climatology, and machine learning, but I am not an expert in deep-time paleoclimate. Thus, I will not comment on those more specific questions and hope that other reviewers can evaluate those aspects more thoroughly.

I explain my main concern below. I recommend that the manuscript be revised, perhaps not major revisions, but I would like to see the revised manuscript.

Main point:

1) The video frame interpolating algorithm FILM has been used as pre-trained without any further fine-tuning with paleoclimate data. Thus, it solely relies on 'video dynamics' that can be found, I assume, in usual video clips and films. These video dynamics are most likely dominated by 'advection' of visual features: static or moving objects, gradual changes in colour, shades, perspectives, etc.

The authors validate the application of FILM in a paleoclimate setting by looking at (1) spatially resolved palaeography and (2) globally averaged temperatures and run-off. This is where my concern arises. The spatially resolved palaeography does resemble a 'video clip'. The movement of continental and ocean plates is indeed an 'convection' feature, and therefore we can expect that a video frame interpolating algorithm can cope

with the interpolation in time of paleography, specially at large continental scales. However, I am not convinced it also works for spatially resolved temperatures. The temperature field is not advected; it does not 'move' like an object in space. Many factors, including land-sea distribution and external forcing, including latitude, CO2 concentrations, water vapour concentrations, precipitation, etc, control it. It is a large leap of faith to consider the evolution of the temperature field as a set of moving objects in space. This might be more correct at very short time scales. Say hours or days, for which temperature may be more strongly controlled by convection of air masses, but this is not true for longer time scales.

The question would be then how to validate the interpolated temperature field. The authors present a validation of the global mean temperature, but as they argue in the manuscript, global mean temperature at these long time scales is strongly controlled by greenhouse gas concentration, and thus, any simple interpolation algorithm would probably achieve satisfactory results without the requirement of a skilful spatially resolved reconstruction. If only the global mean temperature were important, this would be a more or less acceptable validation, but then the FILM setup would not be necessary - a simple time interpolation of the global mean temperature would be sufficient.

I know that a spatially resolved validation is not easy, but why should the FILM output be trusted without that step? One possibility is to use a 'perfect model' approach using GCM simulations. Here, the ground truth is assumed to be a long GCM simulation, which can then be subsampled, interpolated and compared with the 'truth'. I know there are no GCM simulations over such long periods, but some cover the Holocene. Here, the FILM setup can be tested. Alternatively, simulations with intermediate complexity models, like CLIMBER or similar, over longer periods (~100k years ) might be used for this test.

This is an excellent recommendation to test the reliability of our method. We now use a Phanerozoic global surface air temperature dataset (Scotese et al., 2021) from GCM simulations (HadCM3L; Valdes et al., 2021). This dataset shares the same temporal (~5

Myrs) and spatial dimensions (1×1 degree) as the PaleoDEM dataset, and was run using the PaleoDEM data as a boundary condition for the climate model, allowing us to apply identical validation procedures. Our numerical validation results indicate that Deep Learning is effective in interpolating temperature – scoring similarly on most metrics and even better in some than the DEM alone. Likely this is because temperature fields have less abrupt transitions than land-sea masks. We have incorporated this validation into the revised Section 3.2 and thank the reviewer for the suggestion.

3.2 Validation of interpolation using a GCM dataset

We now apply the FILM model to a high-time-resolution dataset of Phanerozoic surface air temperature (SAT; Scotese et al., 2021). This dataset is based on GCM simulations (HadCM3L; Valdes et al., 2021), with the CO2 level in the simulation inferred from global temperature proxies such as biogenic calcite and apatite δ18O and lithological climate indicators. The Phanerozoic SAT dataset shares the same spatial resolution as the PaleoDEM dataset, with a resolution of 1×1 degrees, and comprises a 361×181 data array. The SAT dataset features a 10-Myr temporal resolution from 540-450 Ma and a 5-Myr resolution from 450 Ma to the present. We selected the SAT dataset from 450 Ma onward to ensure a consistent validation.

During validation, we used the SAT dataset without downscaling and conducted the same numerical validation considering temporal intervals of 10 Myrs, 20 Myrs, and 40 Myrs. Similar to the results for the PaleoDEM dataset, interpolations using a 10-Myr interval demonstrated close congruence with the actual frames, as evidenced by high values of SSIM, 2D-correlation, and PSNR, along with low values of NRMSE from 450 Ma to present (see Fig. 5; Table 2). Moreover, compared to the PaleoDEM dataset, the interpolation performance of GMST across different time intervals exhibited more consistent results, as indicated by closer evaluation metrics (Table 2). This is likely because the temperature fields did not contain such sharp transitions between land and ocean.

[Figure]

**Figure 5. Comparative evaluation of performance utilizing (a) 10 Myr, (b) 20 Myr, and (c) 40 Myr intervals within the SAT Dataset.** See Figure 2 for detailed explanations of the image symbols.

**Table 2. Numerical evaluation on the Phanerozoic SAT dataset**

|  | 10 Ma | | | 20 Ma | | | 40 Ma | | |
|---|---|---|---|---|---|---|---|---|---|
|  | Mean | Median | SD | Mean | Median | SD | Mean | Median | SD |
| SSIM | 0.95 | 0.95 | 0.02 | 0.93 | 0.93 | 0.03 | 0.89 | 0.89 | 0.04 |
| PSNR | 32.66 | 32.82 | 2.53 | 31.14 | 30.80 | 2.40 | 29.72 | 29.16 | 2.19 |
| 2D correlation | 0.99 | 0.95 | 0.02 | 0.99 | 0.99 | 0.01 | 0.98 | 0.99 | 0.02 |
| NRMSE | 0.02 | 0.02 | 0.00 | 0.03 | 0.03 | 0.00 | 0.03 | 0.03 | 0.00 |

Particular points

2) 'We then apply the method to upscale the paleoclimate data structure in the SCION climate-biogeochemical model and demonstrate that upscaled outputs for global distributions of surface temperature and runoff follow a logical progression between the original keyframes.'

This sentence is a bit convoluted and not easy to understand. Does it mean that the interpolation produces reasonable or plausible fields?

Yes, that was our intention, we have modified the text.
Lines 20-22: We then apply the method to upscale the paleoclimate data structure in the SCION climate-biogeochemical model. The interpolated surface temperature and runoff are reasonable and present a logical progression between the original keyframes.

3) 'This coarse time resolution likely has impacted the accuracy of the biogeochemical model results'
has likely
 Modified.

4) 'Deep Learning models are complex neural networks with typically >106 parameters'
I guess 106 is a typo. Do you mean 100? Why precisely 106?
Modified. We mean $10^6$ parameters.

5) 'The model emulates the learning process of humans by updating the parameters in the neural networks to produce optimal predictions'

I would not use the term predictions, as the application in this study is not prediction but interpolation. Also, neural networks can generally be used in many other non-predictive settings.

Modified. We use 'optimal results' instead.

6) 'This convolutional operation yields a higher-level representation of the original images'.

The word higher level will not be clear to many readers if they are not experts in machine learning. Can you be more specific?

Modified. We use 'a summarized representation' instead.

7) Table 1. The caption is too cryptic and should not refer the reader to search the text for an explanation of the table's contents. At the very least, it should point to a specific position in the text.

Improved as suggested.

8) The only indication of the time span covered in this paper is the title (Phanerozoic). I think it would be helpful to include a more specific time frame in the abstract and the introduction.

Improved as suggested.

**Added references**

Goddéris, Y., Donnadieu, Y., and Mills, B. J. W.: What models tell us about the evolution of carbon sources and sinks over the Phanerozoic, Annu. Rev. Earth Planet. Sci., 51, 471–492, 2023.

Park, Y., Maffre, P., Godderis, Y., MacDonald, F. A., Anttila, E. S. C., and Swanson-Hysell, N. L.: Emergence of the Southeast Asian islands as a driver for Neogene cooling, Proc. Natl. Acad. Sci. U. S. A., 117, 25319–25326, https://doi.org/10.1073/pnas.2011033117, 2020.

Scotese, C. R., Song, H., Mills, B. J. W., and van der Meer, D. G.: Phanerozoic paleotemperatures: The earth's changing climate during the last 540 million years, Earth-Science Rev., 215, 103503, https://doi.org/10.1016/j.earscirev.2021.103503, 2021.

Valdes, P. J., Scotese, C. R., and Lunt, D. J.: Deep ocean temperatures through time, Clim. Past, 17, 1483–1506, https://doi.org/10.5194/cp-17-1483-2021, 2021.

Wong Hearing, T. W., Pohl, A., Williams, M., Donnadieu, Y., Harvey, T. H. P., Scotese, C. R., Sepulchre, P., Franc, A., and Vandenbroucke, T. R. A.: Quantitative comparison of geological data and model simulations constrains early Cambrian geography and climate, Nat. Commun., 12, 1–11, https://doi.org/10.1038/s41467-021-24141-5, 2021.